# Vastly different energy landscapes of the membrane insertions of monomeric gasdermin D and A3
Viktoria Korn [ORCID] & Kristyna Pluhackova [ORCID] [✉]

Gasdermin D and gasdermin A3 belong to the same family of pore-forming proteins and executors of pyroptosis, a form of programmed cell death. To unveil the process of their pore formation, we examine the energy landscapes upon insertion of the gasdermin D and A3 monomers into a lipid bilayer by extensive atomistic molecular dynamics simulations. We reveal a lower free energy barrier of membrane insertion for gasdermin D than for gasdermin A3 and a preference of gasdermin D for the membrane-inserted and of gasdermin A3 for the membrane-adsorbed state, suggesting that gasdermin D first inserts and then oligomerizes while gasdermin A3 oligomerizes and then inserts. Gasdermin D stabilizes itself in the membrane by forming more salt bridges and pulling phosphatidylethanolamine lipids and more water into the membrane. Gasdermin-lipid interactions support the pore formation. Our findings suggest that both the gasdermin species and the lipid composition modulate gasdermin pore formation.

Gasdermins (GSDMs) are a family of pore-forming proteins conserved among all vertebrates, but even certain bacteria[1–3], fungi[3–5], and invertebrates[3,6] possess gasdermin-like proteins. Gasdermins are categorized into gasdermin A-F and depending on the species, subvariants exist (e.g., mice have three subvariants of GSDMA, named GSDMA1-3, and four of GSDMC, denoted as GSDMC1-4)[7,8]. Although their sequence differs[9,10], gasdermins share the same secondary structure. The only exception is GSDMF, also known as pejvakin or DFNB95, which possesses a truncated C-terminus (UniProt:Q0ZLH3) and whose structure has not been determined experimentally yet. The full-length dormant gasdermin can be cleaved into an N- and C-terminus by various caspases[11].

The N-terminal fragment of gasdermins is a pyroptotic agent (see Fig. 1 for a scheme of pyroptosis execution by gasdermins), which adapts a secondary structure that can be described as a fist (globular domain) with two extended fingers ($\beta$-sheets)[12], that is able to oligomerize into medium-sized membrane pores[13,14] and insert without the help of insertases[11]. Pyroptosis is a form of programmed lytic cell death in which the membrane is perforated, leading to an efflux of cytosolic content, and culminating in ninjurin-1 mediated complete cell rupture[15,16].

Gasdermins are expressed in cells of the gastrointestinal tract and skin[17], hence their name. Their activation takes place downstream of the canonical and non-canonical inflammasomes as a standard response to inflammation and host defence[18], where caspases are responsible for cleavage[19–21]. GSDMD can be cleaved by caspase 1, 4, 5, 8, and 11[22–25].

Recently, more caspase-unrelated cleaving agents have been identified as the streptococcal exotoxin B[26] (cleaving GSDMA), the enterovirus71[27], the zika virus[28], neutrophil elastase[29], and cathepsin-G[30] (all cleaving GSDMD). Up to this day, the cleaving agents of some gasdermins (e.g., GSDMA3) are still unknown.

Overexpression of gasdermins has been linked to many chronic inflammatory diseases like dermatitis and psoriasis[31], as well as cyclic alopecia[32], multiple sclerosis[33], artherosclerosis[34], hyperplasia[35], and hepatocellular carcinoma[36], among others. Thus, gasdermins constitute possible drug targets in diverse disease states[37]. Currently, disulfiram is being investigated as a possible inhibitor of GSDMD pore formation[38], which may be related to C191 palmitoylation aiding GSDMD insertion[39]. Palmitoylation of Cys4 at the globular domain of the bacterial gasdermin from *Vitiosangium* was suggested to mediate protein interactions with the membrane prior to $\beta$-hairpin formation and membrane insertion[2], thus elegantly explaining the ability of bacterial gasdermins to porate membranes of even a simple lipid composition and lacking lipid specificity[1]. Verterbrate gasdermins, on the other hand, specifically target membranes with negatively charged lipids, like phosphoinositides and cardiolipin, but also phosphatidic acid and phosphatidylserine[8,40]. The binding is mediated by a number of positively charged residues on the membrane-exposed interface of the globular domain, i.e., helix $\alpha$1 and strands $\beta$1 and $\beta$2[12,41,42].

This membrane binding specificity of gasdermins is currently being tested as a novel tool to selectively kill cancer cells. Promising results have

Stuttgart Center for Simulation Science, Cluster of Excellence EXC 2075, University of Stuttgart, Stuttgart, Germany.
[✉] e-mail: kristyna.pluhackova@simtech.uni-stuttgart.de

**Fig. 1 | Gasdermins execute pyroptosis. a** The protein is cleaved into the C-terminus (CT, red) and the active N-terminus (NT, blue) (**b**), which forms pores in membranes (**c**) that lead to the efflux of cytosolic components and culminate in cell rupture (**d**). **e** Dysregulation causes disease symptoms.

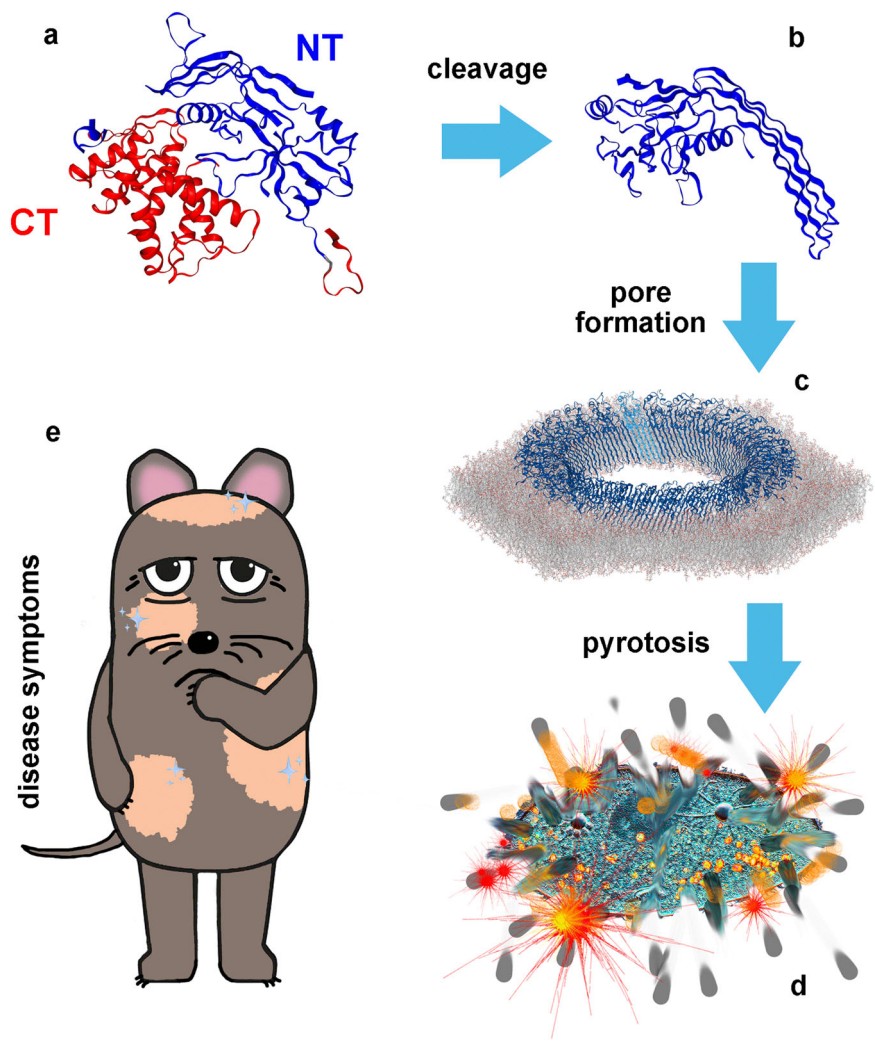

emerged in treating endometrial cancer[43,44], breast cancer, inoperable ovarian cancer, melanoma[45], and colon cancer[46]. GSDMD also protects its host from salmonella infection[47].

The functional studies have been complemented recently by a number of gasdermin pore structures. In 2018, the full pore structure of GSDMA3 was resolved using cryoTEM[41]. In 2021, the structure of GSDMD pore followed[12] and was joined by the GSDMB pore in 2023[48,49]. In 2024, cryo-TEM structures of three bacterial gasdermins[2,50] were discovered. Although detailed structures of a number of GSDM pores and pore-forming oligomers are known, the exact mechanism of pore formation still remains a mystery. For pore-forming proteins localized initially in solution like lysins, perforins, gasdermins, and Bcl-2 proteins[51], two possible main pathways of pore formation have been proposed: The "concerted" pathway in which, after activation, the proteins oligomerize and adsorb to the membrane prior to their simultaneous insertion, and the "non-concerted" one where after activation, monomers adsorb and insert individually before oligomerizing and forming the pore[51]. For visualization see Fig. 2. Pneumolysin follows the concerted mechanism while actinoporin EqtII follows the non-concerted one[51]. For GSDMs, the data from experiments using atomic force microscopy[13,14] and extensive molecular dynamics (MD) simulations[2,14,42,52], is consistent with both suggested pathways.

Despite the conserved overall secondary structure, the sequences of different gasdermins diverge notably, also introducing local distinctions like β-hairpins of different length and polarity. While the different lengths of the transmembrane β-hairpins could be essential for pore formation in membranes of different thickness, their amino acid composition influences interactions and energetics with the lipids and water to a level where they may regulate the type of pore-forming mechanism. Interestingly, gasdermins often carry charged and polar residues on their membrane-inserting β-hairpins. Despite initially thought to be abnormal, over the past years several conserved charged residues have been found on transmembrane domains where they are necessary for function and known to participate in particular interactions[53–56].

Here, using both extensive equilibrium MDs simulations and geometric perturbation ("umbrella sampling"), we investigated the behavior of GSDMD monomers inside, on top of, and on the way through a model *E. coli* lipid bilayer[57]. The comparison of membrane-insertion energetics of monomeric GSDMD to that of GSDMA3[52] reveals drastic differences suggesting that GSDMA3 and GSDMD form pores via different pathways. The different behavior of GSDMD and GSDMA3 monomers is also reflected in the different pore-formation propensity of arcs comprising either seven GSDMD or seven GSDMA3 monomers. Moreover, simulations of monomeric GSDMD in POPC/30% cholesterol support the significant modulatory role of the membrane composition on gasdermin pore formation.

## Methods

The structure of GSDMD in the membrane-inserted state was based on the cryoTEM structure 6VFE[12] with the E192 mutation reversed to L192 and contained residues M1-Q241. The native N-terminus was positively charged, the C-terminus was neutralized by capping via an amine group, and all titratable residues were in their charged state typical at physiological

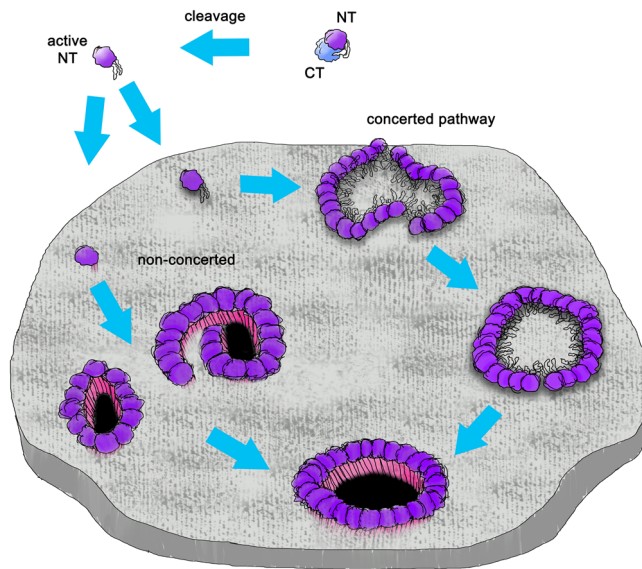

**Fig. 2 | The possible pathways of GSDM pore formation.** Dormant GSDM is cleaved into the C-terminus (CT) and the active N-terminus (NT), which either adsorbs to the membrane, forms oligomers, and inserts ("concerted" pathway), or inserts, oligomerizes in the membrane, and forms the pore ("non-concerted" pathway). Thereby not only full circular pores arise, but also slits and arcs may (temporarily) form. The fist (globular domain of the N-terminus) is shown in purple, and the fingers (β-hairpin-forming residues) are shown as black lines. In the membrane-inserted state, the transmembrane β-sheets are colored in magenta, and the open pore is indicated in black.

pH. In detail, all lysine and arginine residues were positively charged, all aspartic and glutamic acids were negatively charged, histidines were neutral by being protonated by a single hydrogen on the $N_\epsilon$.

## Setup and preequilibration at coarse-grained resolution

Using our established multiscaling procedure[58], a single GSDMD molecule or an arc of seven GSDMDs was inserted into our symmetric model of the *E. coli* polar lipid extract (PLE) membrane[57] to be comparable to our previously performed study of GSDMA3[52]. It is important to note that GSDMD, despite being naturally localized in the plasma or mitochondrial membrane[40,59], can also form pores in *E. coli* PLE[2,13]. Next, GSDMD was coarse-grained using Martini3[60]. Afterward, monomeric GSDMD was surrounded by *E. coli* PLE containing 276 molecules in each leaflet and solvated by 14240 nonpolarizable water beads and 175 $Na^+$ and 22 $Cl^-$ beads by insane[61]. The arc of seven GSDMDs was analogously surrounded by 547 *E. coli* PLE lipids in each leaflet and solvated by 29000 nonpolarizable water beads and 363 $Na^+$ and 54 $Cl^-$. The *E. coli* PLE membrane model consisted of 14 different lipid types: four cardiolipins, five phosphatidylglycerols, and five phosphatidylethanolamines, each including five different lipid tails: palmitoyl (16:0), palmitoleic acid (16:1 $cis^{9,10}$), cis-11,12-octadecenoic-acid (18:1 $cis^{11,12}$), cis-9,10-methylene-hexadecanoic-acid (cy17:0 $cis^{9,10}$), and cis-11,12-methylene-octadecanoic-acid (cy19:0 $cis^{11,12}$)[57]. The lipids were symmetrically distributed between the two membrane leaflets. Additionally, monomeric GSDMD was surrounded by a POPC bilayer containing 30% cholesterol consisting of 177 POPC and 76 cholesterol molecules in each leaflet and solvated by 11000 nonpolarizable water beads, 76 $Na^+$, and 79 $Cl^-$.

The systems were then twice energy minimized. In the first steepest descent energy minimization of 1000 steps, the protein beads were frozen in space. In the second energy minimization of 10,000 steps, all beads were allowed to move. The protein surroundings were then equilibrated using position restraints on the protein in all following steps: Velocities were generated at 310 K, and 5000 steps of a simulation with a timestep of 2 fs were performed. In the second position restraint simulation, a 5 fs timestep

was used, and 10,000 steps were done, followed by 10,000 steps with a 10 fs timestep and by 500,000 steps with a 20 fs timestep. In all those coarse-grained simulations, the electrostatics after a 1.1 nm cutoff were described by the reaction field method with $\epsilon_r = 15$, the van der Waals forces were cutoff at 1.1 nm using the Potential-shift-Verlet modifier[62]. Moreover, the Verlet cutoff scheme[63] with 0.005 buffer tolerance was used, the center of mass of the reference coordinates of the position restraint protein was rescaled, the temperature was kept at 310 K using the v-rescale thermostat[64], a temperature constant of 1 ps for both the "Protein-Membrane" and the "Solvent" bead groups. The pressure was controlled in a semiisotropic manner by the Berendsen barostat[65], using a time constant of 2 ps and a compressibility of $3 \times 10^{-5}\,bar^{-1}$, except for the simulation using a 20 fs timestep, in which a 5 ps time constant of the barostat and a compressibility of $3 \times 10^{-4}\,bar^{-1}$ were utilized.

## Equilibration at all-atom resolution

The preequilibrated systems were converted back to all-atom resolution of CHARMM36m[66,67] using backward[68]. Afterwards, the water was converted to the TIP4P model[69] using a home-written script, and the energy-minimized cryoTEM structure of monomeric GSDMD was fitted to the backwarded protein and replaced it to assure a correct secondary structure of the protein. Overlapping water molecules were removed manually in PyMOL[70], and the system was energy minimized using the steepest descent algorithm in three steps. First, the whole protein was frozen for 5000 steps, then only the protein backbone was frozen for 5000 steps, and in the final energy minimization of 1000 steps, all atoms were allowed to move. Afterwards, velocities were generated at 310 K, and the system was equilibrated by a 50,000 steps protein position restraint simulation using a 0.2 fs timestep followed by 10 ns of a protein position restraint simulation with a 2 fs timestep and 10 ns-long, 2 fs timestep simulation with position restraints on the protein backbone only. In the all-atom simulations, particle-mesh Ewald[71] was used to treat electrostatics beyond the 1.2 nm cutoff, and the van der Waals forces were switched to zero between 0.8 and 1.2 nm using the Potential-switch algorithm. The Verlet cutoff scheme[63] with 0.005 $kJ\,mol^{-1}\,ps^{-1}$ buffer tolerance was used, and the bonds to hydrogens were constrained using the LINCS algorithm[72]. All simulations were performed at 310 K (controlled separately for protein/membrane and the solvent) and 1 bar, using a compressibility of $4.5 \times 10^{-5}\,bar^{-1}$ and semiisotropic pressure coupling. In the position restraint simulations, the Berendsen thermostat and barostat[65] were used with time constants of 0.5 and 5 ps, respectively. In the production run simulations, the Nosé-Hoover thermostat[73] and Parrinello-Rahman barostat[74,75] were used.

After releasing the position restraints, the monomeric GSDMD submerged in *E. coli* PLE was simulated for 5 μs. Afterwards, the β-hairpins of GSDMD were pulled out of the bilayer along the bilayer normal for 2 μs at 2 nm/μs speed. The resulting adsorbed state of the protein was reequilibrated for 1 μs prior to pulling the tips of the β-hairpins (i.e., residues G100 and C191) back into the bilayer for 1.5 μs at 5 nm/μs speed.

The 7mer-arc system was simulated for 4 μs, and the monomeric GSDMD in POPC/30% cholesterol was simulated for 2, 3, and 5 μs after individual equilibration simulations with position restraints on the protein's backbone. All atomistic simulations are listed in Table 1.

## Geometric perturbation

To assure a great diversity of the starting structures for generating the potential of mean force (PMF), 207 snapshots were taken from (i) the equilibrium simulation in the inserted state, (ii) the equilibrium simulation in the adsorbed state, (iii) the pull-out simulation, (iv) the pull-in simulation and, (v) a simulation lasting 500 ns in which the z-distance between the $C_\alpha$ of C191 and the membrane was restrained to 0.5 nm. See Table 1 and Supplementary Fig. 1 and 2 for further details and Supplementary Fig. 3 for exemplary snapshots. The distance along the membrane normal between the $C_\alpha$ of C191 and the membrane middle served as the reaction coordinate $\zeta$. C191 is characteristic for GSDMD and due to its location at the tip of the β-hairpin (Fig. 3) makes the reaction coordinate comparable to that of

GSDMA3, where the distance between K97, also located at the tip of the $\beta$-hairpin, and the membrane middle along the membrane normal was selected as the reaction coordinate[52]. Each geometric perturbation ("umbrella") simulation with a constraining potential using the harmonic force constant of $1000$ kJ nm$^{-2}$ mol$^{-1}$ lasted 50 ns, which was sufficient for convergence (see Supplementary Fig. 4), and the PMF was extracted using the Weighted Histogram Analysis Method (WHAM)[76–78]. The resulting PMF was shifted to zero at $\zeta = 2.5$ nm to allow for a comparison with GSDMA3. All data for GSDMA3 in its charged state (denoted as ccc previously) originates from our recent work[52].

## Analysis
For comparison, we have also analyzed the interactions between the GSDMD monomer and lipid headgroups in the simulations in a model plasma membrane performed by Schäfer et al.[42].

Permeating water molecules were identified by filtering the residue numbers of water oxygen atoms by geometric criteria in each simulation frame. One geometric criterion was the membrane z-dimensions which were estimated for each simulation frame by calculating the center of geometry in z-direction of all lipid atoms, then sorting the oxygen atoms of phosphate headgroups into intracellular and extracellular leaflet, calculating their center of geometry, respectively, and extending the membrane thickness by 2 Å in each z-direction as a buffer to account for an inhomogeneous membrane surface. The second geometric criterion was a cylinder around the center of geometry of the $\beta$-hairpins (residues 86–114 and residues 174–203 for GSDMD and residues 83–109 and residues 167–195 for GSDMA3) with a 10 Å radius and a height of 60 Å. Only water residue numbers appearing within both criteria are then selected and counted.

To determine possible salt bridges on the $\beta$-hairpins, all possible combinations of oppositely charged amino acids were collected in a dictionary. Typically the maximum salt bridge distance (N$^+$ - O$^-$) is defined as 4 Å. Here, in order to speed up the distance calculation, which was conducted for every simulation frame, the atom selection was reduced to one central atom of each charged amino acid type and the salt bridge distance criteria were adapted accordingly: For lysine, the N atom in the terminal amino group was selected, for glutamic acid the C atom of the terminal carboxyl group, for arginine the C atom of the guanidinium group, and for aspartic acid the C atom of the terminal carboxyl group. In the following, the maximum salt bridge distances were defined as 4.58 Å for lysine-glutamic acid, 5.28 Å for arginine-glutamic acid, 5.10 Å for arginine-aspartic acid, and 4.40 Å for lysine-aspartic acid, all based on the trigonometric configurations of the terminal groups.

## Software
All simulations were performed in GROMACS 2021[79], molecules were visualized in PyMOL[70] and nglview[80]. Analyses were conducted using MDAnalysis[81,82], python 3[83], the MDAnalysis-based MembraneCurvature, and GROMACS's tool gmx wham[76]. Plots were generated using Matplotlib[84].

## Results
### GSDMD and GSDMA3 differ in the membrane-inserting $\beta$-hairpin region
The GSDMD monomer possesses longer $\beta$-hairpins than GSDMA3 (see Fig. 4) and a characteristic C191 residue at one tip (Fig. 3) which is known to play an important role during GSDMD pore formation[2,38]. Next to the length also the amino acid sequence of the membrane-inserting $\beta$-hairpins differs significantly. GSDMD carries a manifold of polar and charged residues across the entire hairpins in comparison to GSDMA3 (Fig. 3), which only bears charged residues on its lower and upper end of the hairpins thus allowing them (especially K100 and K102) to snorkel[85–88] to the lipid headgroups[52]. The acidic residues on the $\beta$-hairpins of GSDMD forming acidic patches (AP) 2 and 3 have been proven functionally essential before[12]. In detail, while D87 and E95 at AP2 are important for the specificity of the cargo release, the substitution of all the glutamic acids E169, E171, and E179 at AP3 for alanine compromised GSDMD-mediated liposome leakage of membranes made of cardiolipin, phosphatidylethanolamine lipids, and phosphatidylcholine lipids mixed in 5:8:4 ratio. Notably, E171 and E169 are located above the lipid bilayer and do not interact with lipids in our simulations. Therefore, it is likely that E179 is responsible for the lipid specificity.

As it can be seen in Fig. 4, both proteins form pores of different sizes, the GSDMA3 pore is smaller with its main oligomer consisting of 27

**Table 1 | List of all atomistic simulations of GSDMD**

| Simulation type | Membrane | Description | Length | Sum |
|---|---|---|---|---|
| Equilibrium | *E. coli* | Inserted | 5 µs | 5 µs |
|  |  | Adsorbed | 1 µs | 1 µs |
| Restrained | *E. coli* | Partially inserted | 0.5 µs | 0.5 µs |
| Pulling | *E. coli* | Pull out | 2 µs | 2 µs |
|  |  | Pull in | 1.5 µs | 1.5 µs |
| Umbrellas | *E. coli* | Eq. inserted | 52 × 50 ns | 2.6 µs |
|  |  | Eq. adsorbed | 19 × 50 ns | 0.95 µs |
|  |  | Restrained | 3 × 50 ns | 0.15 µs |
|  |  | Pull out | 91 × 50 ns | 4.55 µs |
|  |  | Pull in | 42 × 50 ns | 2.1 µs |
| Equilibrium | *E. coli* | Inserted 7-arc | 4 µs | 4 µs |
| Equilibrium | POPC/30% cholesterol | Inserted | 3 µs | 3 µs |
|  |  | Inserted | 5 µs | 5 µs |
|  |  | Inserted | 2 µs | 2 µs |
| All |  |  |  | 34.35 µs |

Unless stated otherwise, GSDMD was studied in monomeric form.

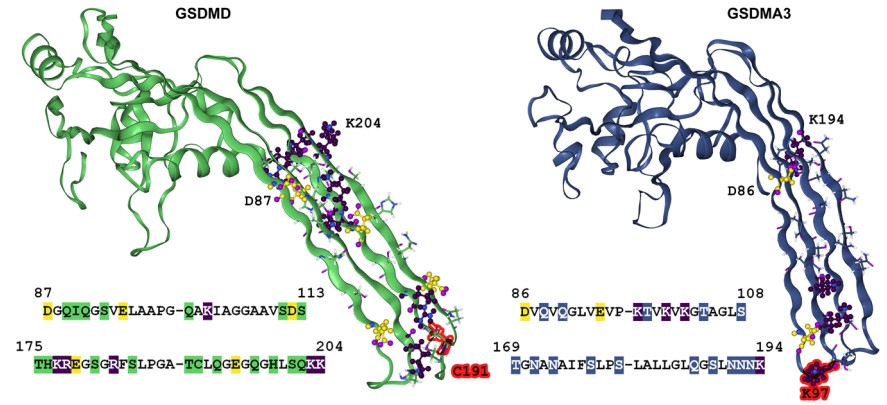

**Fig. 3 | CryoTEM structures of GSDMD (green) (6VFE) and GSDMA3 (navy) (6CB8) monomers in comparison.** On the membrane-inserting $\beta$-hairpins, polar residues are shown as cartoon, and charged residues are shown as ball&stick and colored purple (positively charged lysine and arginine) and yellow (negatively charged aspartic and glutamic acid) together with their amino-acid sequences. Highlighted in red are C191 (GSDMD) and K97 (GSDMA3), whose position relative to the membrane's center of mass constitutes the reaction coordinate for geometric perturbation simulations.

**Fig. 4 | Comparison of the pore structures and monomer conformations within the pore between GSDMD and GSDMA3. a** CryoTEM structures of the 33-mer GSDMD (green) (6VFE) and 27-mer GSDMA3 (navy) (6CB8) pores. **b** Pore monomers overlaid on the globular domains to emphasize the different orientations and lengths of the β-hairpins.

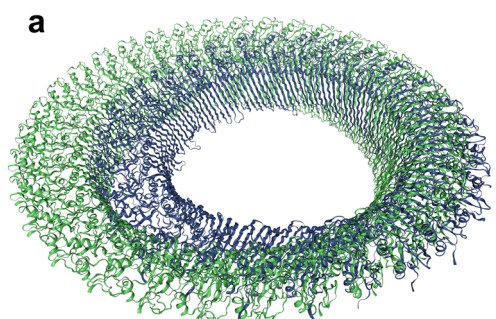
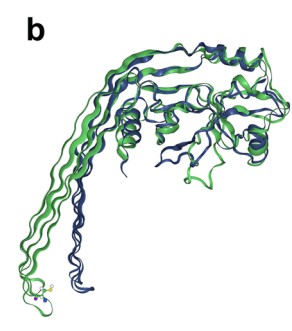

monomers[41], while the GSDMD pore usually consists of 33 units[12]. For both proteins, slit-shaped, elliptic, and round pores with different numbers of subunits (D: 16, 20, 30[13], 26 to 28[12]; A3: 18, 21, 27, 30[14], 31–34[41]) have been reported. Lipid-filled gasdermin oligomers or oligomers adsorbed to the membrane surface are typically denoted as pre-pores which may be of different shapes (arc, slit, ring)[13,14]. Notably, the membrane's lipid composition has been proven to influence both the pore size and shape[2,89].

Primary protein structure is known to influence the energetics of the monomer insertion due to chemical interactions between the protein and membrane lipids as well as water molecules. In our previous research[52], we observed that GSDMA3 prefers the membrane-adsorbed state over the membrane-inserted state in its lowest energy state, in which the aspartic acid and lysine residues at the β-hairpins are charged. The PMF calculations revealed the GSDMA3 monomer has to surpass a 5.6 kcal mol$^{-1}$ barrier on its way into the membrane, although mediating mechanisms like snorkeling, lipid bending, salt bridge establishment, 'piggy-back', and water defect formation are already in place, drastically reducing the energy barrier compared to the membrane passage of individual amino acids or their side chain analogs[52,90,91]. In the full pores of GSDMD and A3, the inside is more hydrophilic, and the outside, facing the lipids, more hydrophobic[41], similar to other β-barrel forming proteins[92,93]. Acidic patches line the inside of the pore and basic ones the outside facing the lipid headgroups. Thereby GSDMD exhibits stronger electrostatics at its surface[12,41]. Thus, based on the primary protein structure of the GSDMD monomer, one could expect a higher barrier due to the longer β-hairpins that bear more hydrophilic and charged residues than for GSDMA3, as inserting charged residues into a hydrophobic membrane core comes at a high energy cost[90,94].

## Membrane-inserted GSDMD and GSDMA3 adapt different conformations

In the here performed equilibrium simulations (Table 1), the GSDMD monomer stayed stably inserted inside the *E. coli* lipid bilayer for multiple, in detail five, μs. The β-hairpins twisted around the water molecules in comparison to the pore cryoTEM structure where the β-sheets are stabilized in a flat shape by hydrogen bonding to the neighboring monomers and β-barrel formation (see Fig. 5c and Supplementary Fig. 5b). This behavior is also reflected in the RMSD (see Supplementary Fig. 6), where the hairpins' RMSD deviates from the pore structure by 0.2–0.6 nm, with maximum values reached for inserted GSDMD at around 1 μs. As the secondary structure analysis (see Supplementary Fig. 7) does not show any significant loss of β-sheet residues, these increased RMSD values reflect the reorientation of the whole hairpins rather than their unfolding. The globular domain remains stable both in the inserted and the adsorbed state (Supplementary Fig. 6a, b).

In contrast, as observed in our previous simulations[52], the GSDMA3 monomer transfers into the cytosolic lipid layer with the β-hairpins extended forward (see Supplementary Fig. 5d) and forms defects that contain fewer water molecules (see Fig. 5). Interestingly, not only water molecules but also phosphate headgroups enter the hydrophobic membrane core and stabilize the proteins inside the membrane. Our analysis revealed the presence of twice as many lipid phosphates in the membrane core in the

GSDMD system than in the GSDMA3 system (Fig. 5b). Such deformations have been observed before, especially if arginines are involved[95]. A more detailed view unveiled that especially phosphatidylethanolamine (PE) lipids interact by their positively charged amino groups with negatively charged residues on the β-hairpins of GSDMD (see Fig. 6).

These interactions and the resulting formation of a membrane rim enable the GSDMD monomer to pull roughly twice as many water molecules into the membrane than the GSDMA3 monomer. The similarity of our observations of the GSDMD monomer in the *E. coli* lipid extract to those of GSDMD in a model of the plasma membrane[42], including PE lipid headgroups inside the membrane (visualized in Supplementary Fig. 8), show that the GSDMD monomer behaves similarly in both membrane types and that PE lipids specifically bind to the β-hairpins of GSDMD. PE lipids are known to induce negative curvature stress and modulate the lateral pressure profile of membranes which can enhance the formation of pore-like defects and promote the partitioning of polypeptides into the membrane interface[96,97]. Analysis of membrane curving caused by inserted monomeric GSDMD and GSDMA3 (shown in Supplementary Fig. 9) indeed shows larger curvature around β-hairpins of GSDMD than GSDMA3. The importance of those specific protein-PE interactions is confirmed by our all-atom simulations of monomeric GSDMD inserted in a simple plasma membrane mimic bilayer consisting of POPC and 30% cholesterol. The lack of PE/β-hairpin interactions destabilizes the membrane-inserted state of GSDMD, resulting in rapid repulsion of the β-hairpins from the membrane in two simulations and partial excision in the third simulation (see Supplementary Figs. 10 and 11). The above-observed differences of monomeric GSDMD and GSDMA3 in their ability to form water-filled pores in the *E. coli* membrane get even more pronounced if multiple proteins are membrane inserted as shown in our simulations of membrane-inserted arcs consisting either of seven GSDMD or seven GSDMA3 monomers (see Fig. 7). While GSDMA3 maintains its arc shape even after 4 μs of simulation time, the GSDMD oligomer pulls water molecules and lipid headgroups to its β-hairpins and wraps around forming an elliptic water-filled pore, similarly to its behavior in the plasma membrane observed earlier by Schäfer et al.[42].

## The strikingly different energetics of membrane insertion/excision of GSDMD and GSDMA3

Based on simulations in which we pulled GSDMD into and out of the membrane, equilibrium simulations, and the geometric perturbation simulations derived therefrom, we could observe a PMF profile strikingly different to that of GSDMA3 we have estimated recently[52]. For comparison of both PMFs see Fig. 8d.

Keeping in mind the limited structural sampling (folding/unfolding) of the β-hairpins of both GSDMD and GSDMA3 in the cytosol, the comparison of the simulations shows that the membrane-adsorbed state of GSDMD is not characterized by a single potential well as in the case of GSDMA3 but comprises multiple small potential wells. Together with the comparatively large error bars of the PMF in this region, GSDMD obviously does not have a single well-defined and stable adsorbed state. Visually, GSDMD can be observed drilling its hairpins a little deeper into the

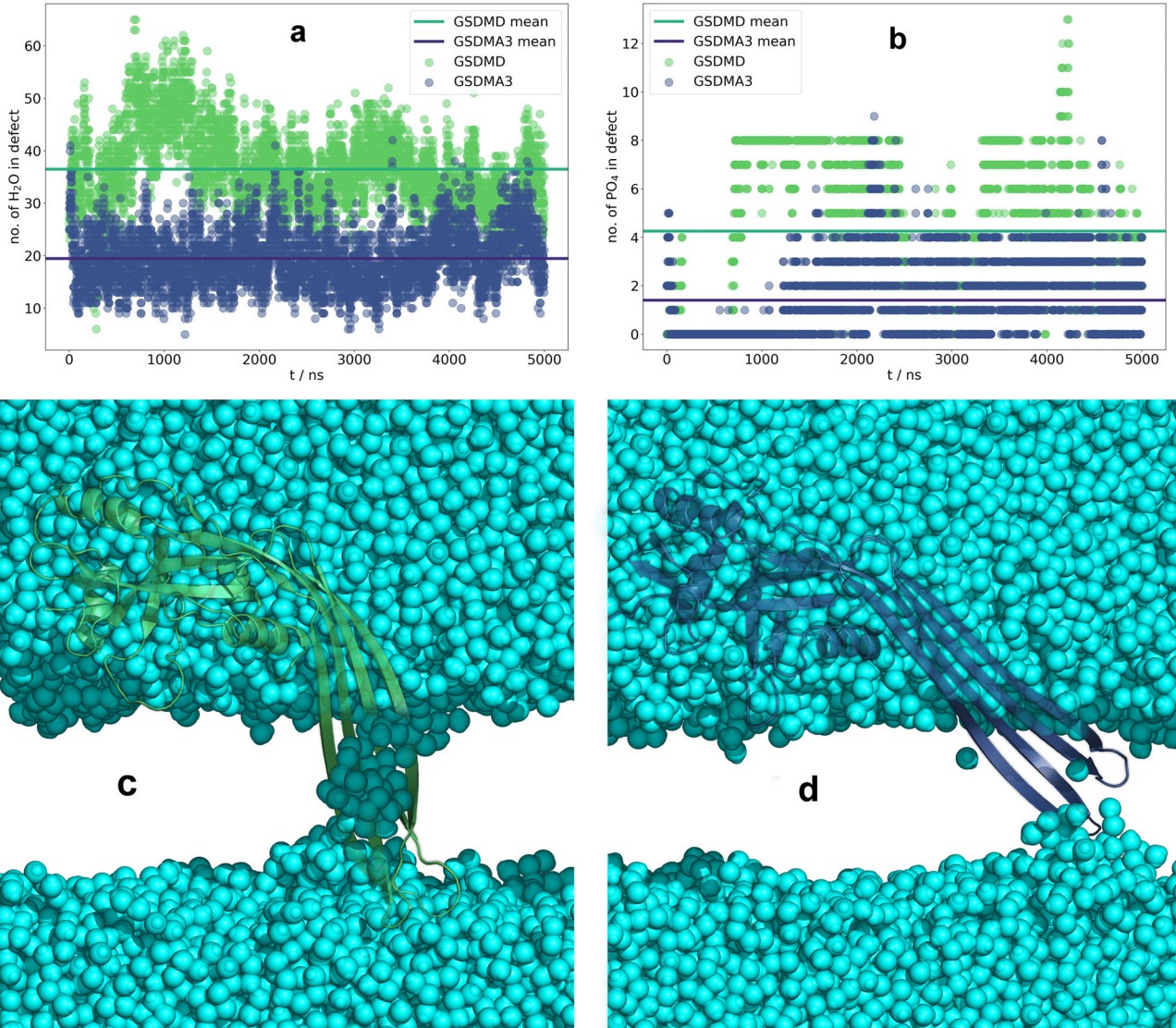

**Fig. 5 | Membrane defect formation by the GSDMD and GSDMA3 monomers in 5 µs long equilibrium simulations. a** GSDMD (green) forms significantly larger water defects than GSDMA3 (navy), which is likely rooted in the higher polarity of GSDMD's β-hairpins and in the structures the monomers adapt inside the bilayer (shown in **c, d** and in Supplementary Fig. 5). **b** Both proteins pull phosphate headgroups into the membrane defects. In the GSDMD simulation, about twice as many lipid headgroups are pulled into the defects. In the inserted state GSDMD establishes a continuous water pore, while GSDMA3 does not (**c, d**). The proteins in **c, d** are shown as green (GSDMD) or navy (GSDMA3) cartoon, and the water molecules as cyan spheres.

membrane, presumably due to its more hydrophilic surface that is able to interact with the lipid headgroups, and already drags water molecules along at higher values of $\zeta$ than GSDMA3 (Fig. 8e). On its way into the membrane, GSDMD meets a much smaller barrier of only 2.0 kcal mol$^{-1}$, compared to 5.6 kcal mol$^{-1}$ observed for GSDMA3, and settles deeper inside the bilayer, namely at a $\zeta$ of ~−1.50 nm, compared to −0.66 nm for GSDMA3. While GSDMA3 strongly deforms the membrane below the flatly inserted β-hairpins at the barrier, GSDMD accumulates water molecules on its hairpins and is thereby able to insert them deeper (Fig. 8). The energy minimum of GSDMD in the membrane-embedded state is broader, and the potential well is 5.9 kcal mol$^{-1}$ deep, compared to 4.4 kcal mol$^{-1}$ of GSDMA3. In this state, both proteins twist their hairpins to stabilize the membrane defect and accommodate water molecules. It is clearly visible how GSDMD profits from its longer, highly polar hairpins as it inserts them deeper, already reaching the extracellular headgroup region and forming an almost entirely water-filled pore (Fig. 8, bottom left). The lipid headgroups being pulled into the membrane follow the same trend as the water molecules: GSDMD interacts with more lipid headgroups already when it starts inserting and

maintains more contacts during full insertion (Supplementary Fig. 12). Altogether, the smaller energy barrier for membrane insertion and the deeper potential well in the membrane-inserted state result in 3.9 kcal mol$^{-1}$ preference of monomeric GSDMD to be in membrane-inserted state compared to the membrane-adsorbed state. This behavior is completely opposite to that of monomeric GSDMA3 which by 1.1 kcal mol$^{-1}$ prefers the membrane-adsorbed over the membrane-inserted state. Knowing that membrane insertion of polar and charged amino acids is energetically very costly[90,91,98,99], we started searching for specific reasons for GSDMD's ability to swiftly insert into the membrane's hydrophobic core and comfortably stay there. Previously it has been observed that a sufficient amount of hydrophobic residues can counterbalance polar or charged residues on transmembrane helices[100]. For GSDMA3, this might be the case, but for GSDMD, such a mechanism is unlikely. The same study also identified protein context in membranes responsible for lower insertion costs of charged residues, which in our case is not applicable, either. Yet, membrane inhomogeneity due to water defects can be a major driving force in facilitated membrane insertion of charged and polar species. Yao et al.[101] analyzed

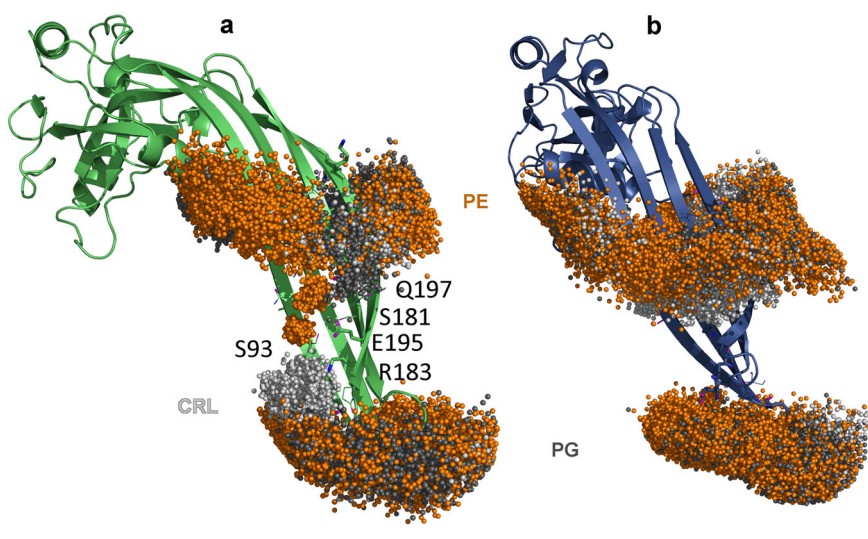

**Fig. 6 | Interactions of lipid headgroups with the GSDMs.** Localization of the phosphorus atoms within one nm of the β-hairpins of GSDMD (**a**, shown as green cartoon) and GSDMA3 (**b**, shown as navy cartoon) in the last 1 μs of the 5 μs long simulations. The phosphorus atoms are visualized as spheres and colored according to the lipid type: Phosphatidyletanolamine phosphates (PE) are colored orange, cardiolipin phosphates (CRL) are colored light gray and phosphatidylglycerol phosphates (PG) are colored dark gray. In GSDMD, charged and polar residues mediating the protein-lipid interactions are shown as sticks and lines, respectively. Q197, S181, E195, R183, and S93 in the middle of GSDMD's β-hairpins are labeled, as they, together with E179 (obscured behind the phosphorus atoms), specifically stabilize phosphatidy-lethanolamine headgroups (orange spheres) and cardiolipins (light gray) deep in the membrane hydrophobic core.

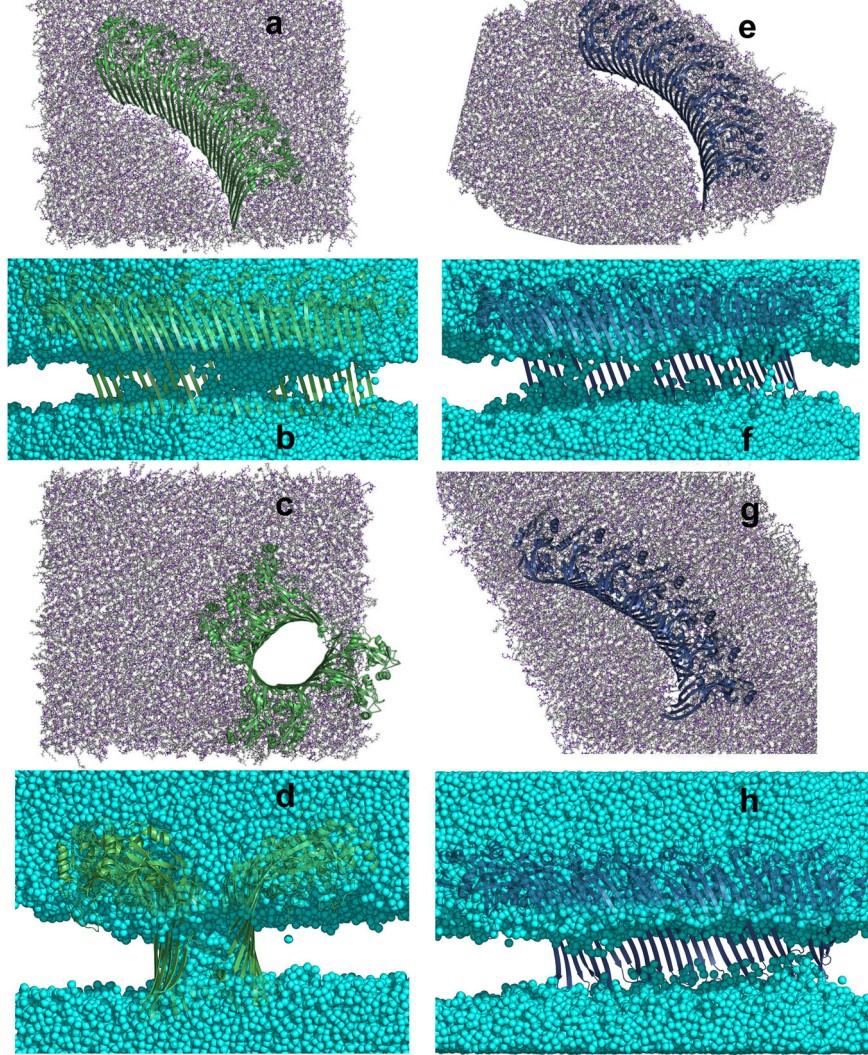

**Fig. 7 | Simulations of arc-shaped oligomers of GSDMD and GSDMA3.** Distinct behavior of GSDMD (green, **a–d**) and GSDMA3 (navy, **e–h**) arc-shaped oligomers comprising seven monomers at the beginning (**a, b, e, f**) of and after (**c, d, g, h**) 4 μs simulation time in an *E. coli* membrane (shown as gray-purple licorice). The GSDMD oligomer forms a water-filled (cyan spheres) pore (**c, d**), while GSDMA3 maintains its arc shape. The simulation of the GSDMA3 arc was taken from our recent work[52]. **a, c, e, g** the proteins (shown as a cartoon) and lipids (shown as licorice) are shown from the cytosolic view, and the water is omitted for clarity. In the side views shown in **b, d, f**, and **h**, the lipids are omitted from the visualization, and the water is shown as cyan spheres.

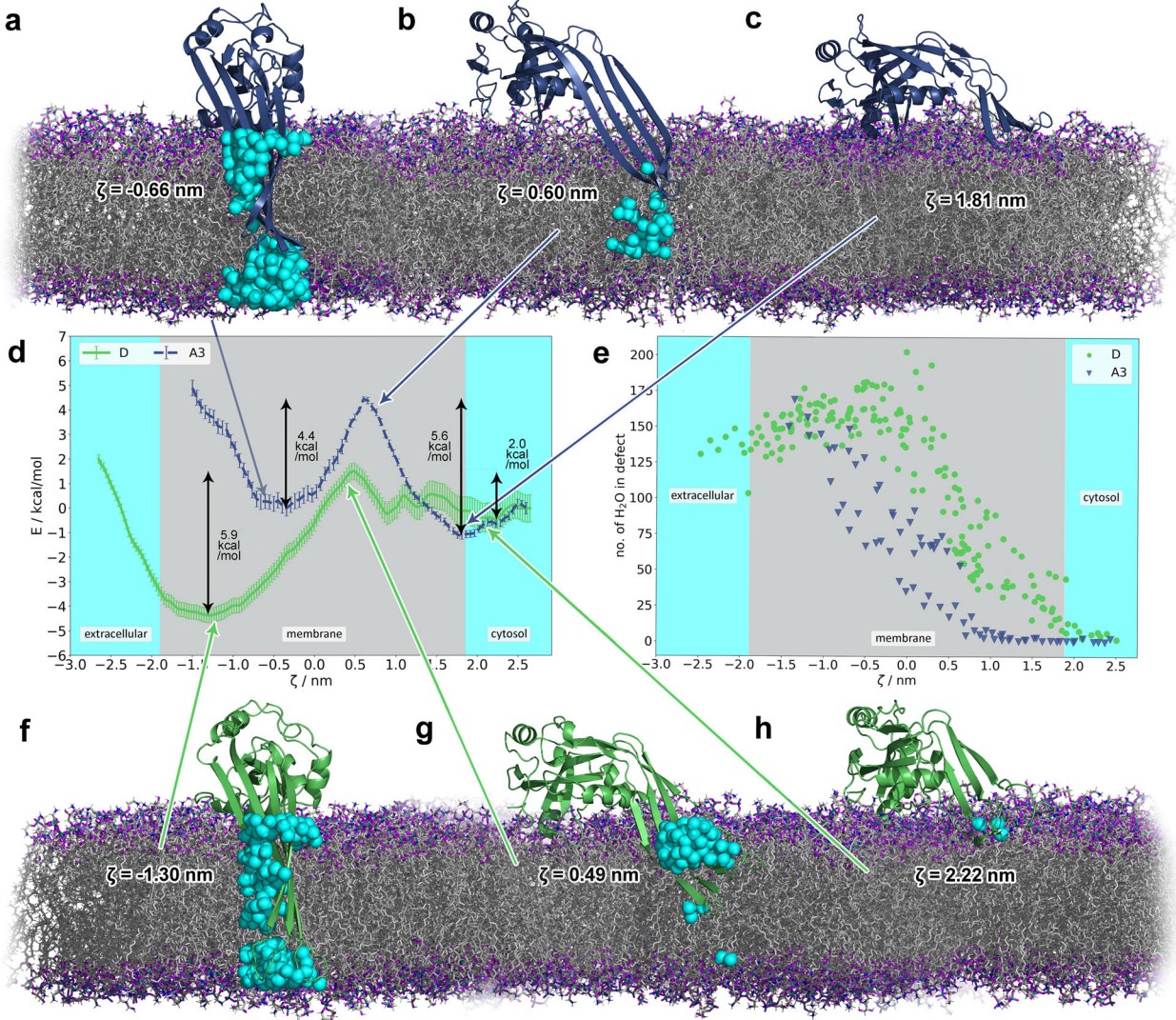

**Fig. 8 | Insertion/excision behavior of GSDMD and GSDMA3.** Potentials of mean force for the GSDMD (green) and GSDMA3 (navy) monomers (**d**) in *E. coli* membrane. The reaction coordinate $\zeta$ is defined as the distance between the $C_\alpha$ (C191 (GSDMD), or K97 (GSDMA3)) and the center of mass of the membrane along the membrane normal. The error bars of the PMF curves denote standard deviations estimated over 4 random parts of the trajectories. **e** Number of water molecules in the water defects over the course of the geometric perturbation simulations. **a–c** and **f–h** Snapshots of GSDMA3 (**a–c**) and GSDMD (**f–h**) in the inserted states (**a, f**), at the barriers (**b, g**) and in the adsorbed states (**c, h**). The proteins are shown as cartoons, GSDMD is green and GSDMA3 is navy, *E. coli* lipids are shown as gray sticks with purple oxygens and blue nitrogens in the headgroup region, the water molecules in the membrane defects are shown as cyan spheres, other water molecules and ions were omitted for clarity. The intracellular space is above, and the extracellular space is beneath the bilayers.

in great detail the free energy contributions to the total PMF of the membrane passage of a peptide comprised solely of nine arginine residues and stated that guanidinium-phosphate salt bridges account for 65% of the overall interaction energy, and generally salt bridges and water-filled pores decrease the free energy barrier of insertion.

The ability of GSDMD to form water-filled membrane defects during geometric perturbation (Fig. 8e) is consistent with the above-described equilibrium simulations (Fig. 5). In detail, GSDMD is able to pull roughly twice as much waters into the membrane as GSDMA3 at the same $\zeta$. Lipid headgroups interacting with the protein follow the same trend (see Supplementary Fig. S10). Even in the adsorbed state of GSDMD, some water molecules are dragged into the membrane by the $\beta$-hairpins' hydrophilic surface ($\zeta = 2.22$ nm, Fig. 8h), facilitating the insertion. The proteins' "fingers" form a shovel-like structure, which in the case of GSDMD, is filled with water, while GSDMA3's "shovel" ($\zeta = 1.81$ nm, Fig. 8c) is empty. In the same region, GSDMD already induces membrane deformations and interacts with lipid headgroups, while GSDMA3 does so the earliest at its own barrier. At the barriers ($\zeta = 0.60$ nm, GSDMA3, and $\zeta = 0.49$ nm, GSDMD), where

the proteins are the most "uncomfortable", the membrane defects are opened, that is energetically unfavorable[102]. Water enters the bilayer from both sides, in the case of GSDMD more numerously. As lipid headgroups line the inside of the defect, the barrier of GSDMD is lower compared to GSDMA3, whose polar residues and water molecules it drags with interact with the hydrophobic lipid tails. In this region of the reaction coordinate, GSDMD pulls the most water into the membrane (Fig. 8e). When settling in the membrane-inserted state ($\zeta = -0.66$ nm, A3, and $\zeta = -1.30$ nm, D), the monomers twist their "fingers" to better accommodate the water molecules inside the membrane defects. While GSDMD maintains a number of lipid headgroups in the defect, GSDMA3 releases the lipid headgroups from the defect when approaching the membrane-inserted state.

As salt bridge formation is an important mechanism shielding off charges on the protein surface from the membrane environment[52,103–106], we have compared the salt bridge formation capability of GSDMA3 and GSDMD over the course of the geometric perturbation simulations. While GSDMA3 possesses just four charged amino acids at the $\beta$-hairpins that are spatially able to form salt bridges (E94, K97, K100, and K102) and only two

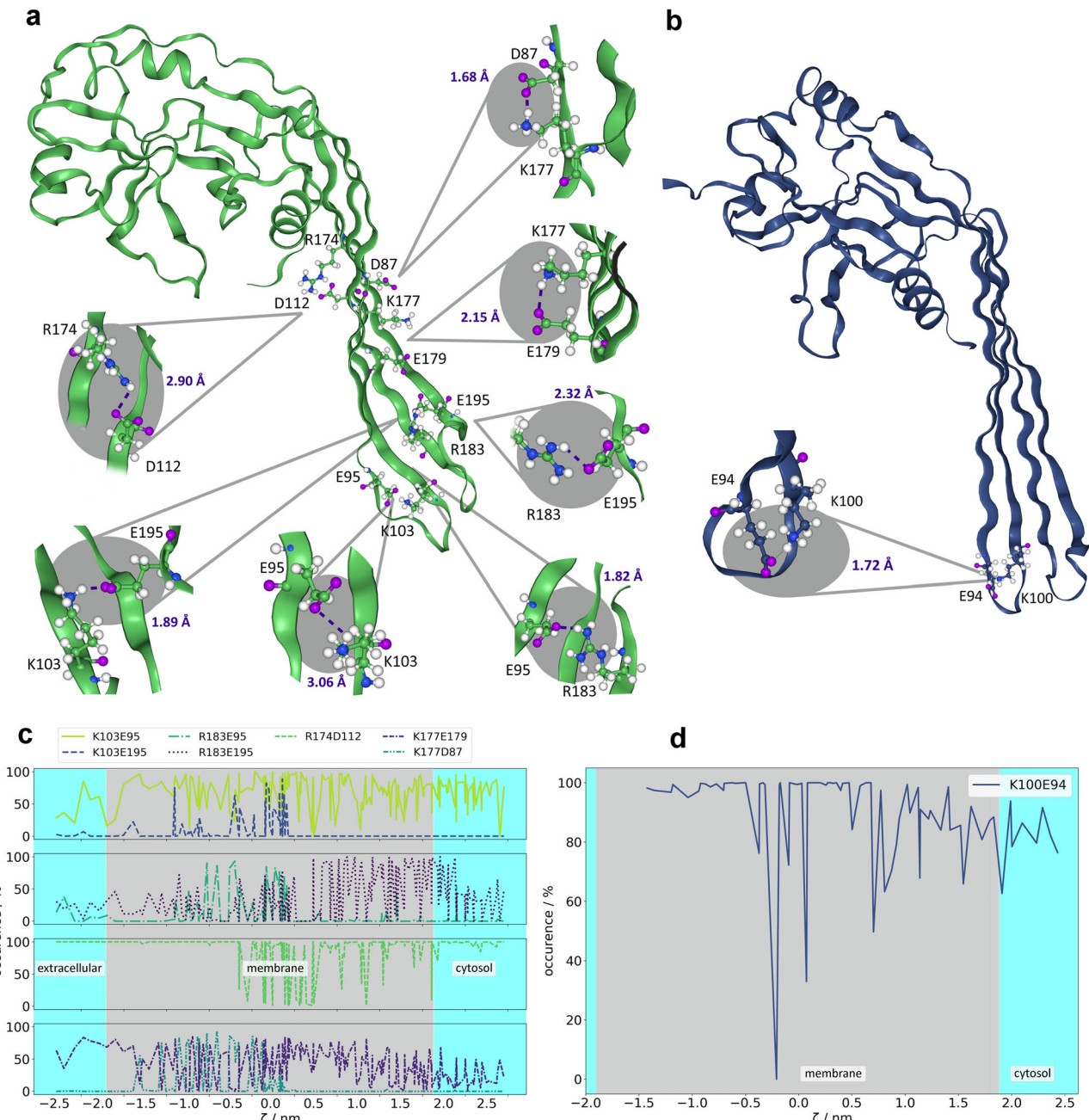

**Fig. 9 | Salt bridge formation of the GSDMD and GSDMA3 monomers.**
**a, b** Locations of salt bridge-forming charged amino acids on the β-hairpins of GSDMD (**a**) and GSDMA3 (**b**) with detailed snapshots from the geometric perturbation simulations. The purple values give the actual distances indicated as dashed lines in the given snapshot. **c, d** Occurrence of salt bridges over the course of

the geometric perturbation simulations for GSDMD (**c**) and GSDMA3 (**d**). The reaction coordinate ζ is defined as the distance between the $C_{\alpha}$ (C191 (GSDMD), K97 (GSDMA3)) and the center of mass of the *E. coli* membrane along the membrane normal.

of them do actually form one (K100E94, see Fig. 9b, d), GSDMD carries a number of charged residues on its β-hairpins, theoretically allowing the formation of 18 salt bridges, of which seven are actually formed in our simulations (Fig. 9a, c). Exemplary time courses are shown in Supplementary Figs. 13–15. K103 mostly interacts with E95 (69.7% of total simulation time), and sometimes it gets closer to E195 (5.6% of total simulation time) (at $-1.2 < \zeta < 0.3$), which mostly interacts with R183 (39.7% of total simulation time). Sometimes R183 can be seen switching over to E95 (8.9% of total simulation time) (mostly between $\zeta = -1.2$ and 0.3), so K103 and R183 share their partners. R174 mostly interacts with D112 (87.4% of total simulation time), forming the most stable and continuous salt bridge. K177

often forms a salt bridge with E179 (39.2% of total simulation time). Only at full insertion, it briefly interacts with D87 (7.8% of total simulation time) (at $-1.7 < \zeta < 0.2$), too. Generally, the salt bridges are present in the cytosol, too, and fluctuate more inside the membrane. During membrane insertion, at the barrier of the PMF, the protein attempts to shield off the charges as well as possible, recruiting all available resources (including all available amino acids) to form salt bridges. The comparatively low involvement of E179 (39.2% of total simulation time) and E195 (45.3% of total simulation time) in intramolecular salt bridge formation, despite their position at the bilayer center results from their interactions with the positively charged amino groups of the PE lipids (Fig. 6).

## Discussion

Although GSDMD and GSDMA3 belong to the same protein family of pore-forming executioners of pyroptosis, these proteins differ in many ways, starting from the primary protein structure over the cleaving agents, preferred membranes to riddle, size, and shape of the pore-forming oligomers and pore-forming dynamics. In this manuscript, we have compared the behavior of GSDMD and GSDMA3 monomers in the same *E. coli* polar lipid extract bilayer in both equilibrium and pulling simulations as well as geometric perturbation ("umbrella sampling") simulations in order to unveil the energy differences in the membrane-inserted state, upon membrane passage, and attached to the membrane surface. Strikingly, despite GSDMD possessing longer $\beta$-hairpins with way more charged amino acids than GSDMA3, it faces a much smaller insertion barrier and prefers the membrane-inserted over the membrane-adsorbed state, which is completely opposite to the behavior of GSDMA3[52]. Numerous mediating effects lower the insertion barrier of GSDMD, like water defect formation, "piggy-back", interactions with lipid headgroups, and salt bridge formation, with the latter two being the most striking in comparison to GSDMA3 due to their abundance. GSDMD prefers the inserted state by 3.9 kcal mol$^{-1}$ (in contrast to GSDMA3), it settles deeper inside the membrane by about one nanometer, forms larger water defects, and has a much more pronounced potential well in the membrane-inserted state (5.9 kcal mol$^{-1}$ vs. 4.4 kcal mol$^{-1}$). The membrane-adsorbed state of GSDMD is diffuse with several shallow minima and is separated by an energy barrier of just 2.0 kcal mol$^{-1}$ from the membrane-inserted state. We were able to study the interactions of the charged amino acids in detail, proving the formation of seven salt bridges that shield off the charges during the membrane insertion of GSDMD. At the same time, the negatively charged residues on the $\beta$-hairpins of GSDMD (E179 and E195) specifically attract the headgroups of positively charged phosphatidylethanolamine lipids, leading to a membrane rim formation surrounding the water-filled membrane pore on their membrane-facing side. A posteriori analysis of MD simulations of monomeric GSDMD in a model plasma membrane performed before[42] shows the same attraction and membrane insertion of positively charged phosphatidylethanolamine lipids by negatively charged residues on the $\beta$-hairpins, supporting the specificity of the interaction and the role of phosphatidylethanolamines in GSDMD membrane incorporation. Similar interactions have previously been observed for YidC, which is an *E. coli* native insertase, where a charged residue 366 (R in the native state, but its mutation to E is also fully functional) interacts with the lipid phosphates, creates water-filled cavities, thins the lipid membrane, and thus eases the insertion of other proteins by this insertase[107–109]. In the light-harvesting complex II, residue E20 is responsible for pulling PE lipids into the membrane and facilitating insertion[96] and GSDMD's E179 and E195 may behave similarly. In a study of the polybia-MP1 peptide, which perforates membranes of distinct lipid composition and possesses charged residues on its transmembrane domain, PS and especially PE lipids dramatically enhanced the formation of membrane defects[97]. The here observed into-membrane attraction of PE lipids by E179 and E195 and the resulting pore formation could contribute to the experimentally observed impairment of liposome leakage by GSDMD containing a triple mutation of negatively charged residues E169A, E171A, and E179E located in the AP3[12]. Additionally, the importance of complex lipid composition for successful membrane leakage has been demonstrated experimentally before[11]. In detail, while only 50% leakage could be observed for liposomes made of 80% PC and 20% PIP2, liposomes made of 45% PC, 35% PE, 5% PS, 5% PI, and 10% PIP2 showed 100% leakage. In order to further elaborate on the role of complex lipid composition for membrane pore formation, we have performed additional simulations of monomeric GSDMD inserted in a model membrane consisting of POPC with 30% cholesterol. Strikingly, the $\beta$-hairpins of GSDMD slipped out of the bilayer within a few µs in two out of three simulations (see Supplementary Figs. 10 and 11), proving the modulatory role of specific lipids in stabilizing GSDMD in its membrane-inserted state. Moreover, the differences between GSDMD and GSDMA3 monomers

concerning the formation of water-filled membrane defects and pores are a foundation for the vastly different behavior of their larger oligomers. Here we have studied arcs comprised of seven monomers. While the arc of GSDMA3 stays stable over the course of the simulation and causes only small water defects, the arc of GSDMD pulls water and lipid headgroups to its $\beta$-hairpins, forming a free membrane edge. The generated edge tension of the membrane rim pulls the terminal monomers of GSDMD together, wrapping it around the water defect, and forming a water-filled transmembrane pore, similar to the behavior of GSDMD in the plasma membrane model[42].

Taken together, based on our simulations, we contribute to filling the knowledge gap on the molecular processes between the cleavage of GSDMs and the formation of their full transmembrane pores and consolidate the two suggested mechanisms of GSDM pore formation, namely membrane insertion of GSDM monomers followed by pore assembly in the membrane and prepore assembly in solution or on the membrane surface followed by synchronized membrane insertion of GSDM oligomers. In detail, our results suggest that depending on the GSDM's properties, a given GSDM may prefer one of the postulated mechanisms of pore formation over the other. At the same time, as our simulations show, protein-lipid interactions play a crucial modulatory role in the process.

## Data availability

The force field files, jupyter notebooks for analysis, and snapshots and trajectories of GSDMD simulations are available online at DaRuS at https://doi.org/10.18419/darus-4474. The numerical data used for the plots is included in the Supplementary Data 1 below.

## Code availability

The analysis and evaluation code are available as jupyter notebooks at DaRus at https://doi.org/10.18419/darus-4474.

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

## Acknowledgements

We thank Siewert-Jan Marrink and Paulo Cesar Telles de Souza for sharing their Martini3.0.4 force field version and the cholesterol parameters with us. Moreover we thank Daniel J. Müller, Stefania Mari, and Yu Han (ETH Zürich) for fruitful discussions on gasdermin pore formation. The authors acknowledge the support by the Deutsche Forschungsgemeinschaft under Germany's Excellence Strategy–EXC 2075–390740016 and by the Stuttgart Center for Simulation Science (SC SimTech). The authors gratefully acknowledge the scientific support and HPC resources provided by the Erlangen National High Performance Computing Center (NHR@FAU) of the Friedrich-Alexander-Universität Erlangen-Nürnberg (FAU) under the NHR project MoTrNanoMat. NHR funding is provided by federal and Bavarian state authorities. NHR@FAU hardware is partially funded by the German Research Foundation (DFG)–440719683. Some of the simulations were also performed on BwForCluster BiNAC funded by the Ministry of Science, Research and the Arts Baden-Württemberg and by the Federal Ministry of Education and Research.

## Author contributions

K.P. conceptualized the research and both authors performed and analyzed the simulations, interpreted the results, and wrote the article.

## Competing interests

The authors declare no competing interests. K.P. is an Editorial Board Member for Communications Chemistry but was not involved in the editorial review of or the decision to publish this article.
