## [Transparent Peer Review file · Communications Chemistry]

Vastly different energy landscapes of the membrane insertions of monomeric gasdermin D and A3

Corresponding Author: Dr Kristyna Pluhackova

Version 0:

Reviewer comments:

Reviewer #1

(Remarks to the Author)

The paper by Korn and Pluhackova is a molecular dynamics study of GSDMD pore formation. The authors compare their results with previous simulations of GSDMA3 simulations. Standard MD protocols have been used in their study and the authors have some potentially interesting conclusions. Overall this is an interesting study. There are several points that the authors need to address and clarify before the manuscript can be considered for publication. The main issue is related to the umbrella sampling simulations where the sampling time at each window is only 50 ns. Apart from this there are several points that the authors need to address and revise the manuscript before it can be considered for publication. The results are simply written in one continuous text and the overall presentation has to be considerably improved.

1. The introduction reads as one long paragraph. The authors need to split this into separate paragraphs and provide some pictures/schematics for the paper to be more accessible for the non-specialist.
2. Figure 2: The numbers of the end residues should be marked in the respective CryoTEM structures for the reader to follow the sequence in the structure. The authors have superimposed the two structures to depict the pore. This is confusing. A separate and improved figure is needed to depict the pore for GSDMD along with the superposition to depict the differences.
3. These pores are usually formed with a range of monomers. The pore size being determined by the protein concentration, membrane composition etc.. This aspect should be mentioned unless there is clear evidence to the contrary.
4. It was not clear how the water channel is stabilized when the pore forms. Typically for the beta-toxins from the family of cholesterol dependent cytolysins the hydrophilic regions form the water channel with the hydrophobic regions facing the lipids. Is there any similarity in this trend for the proteins studied here. If so this should be mentioned somewhere to draw a connect. Hence it is not entirely obvious if longer beta-hairpins should lead to a higher barrier for entry.
5. A table of the different simulations carried out should be mentioned in the methods section. Again this section should be split up into the restraint simulations and the umbrella sampling simulations. In the Results and Discussion
6. In the methods section the authors mention that they carry out simulations with the membrane bound states. However this data is not shown. It would be useful to comment on the differences observed for the two proteins when compared with the membrane inserted states. This would complement the membrane inserted states show in Figure 2. These simulations will also reveal the stability fo the pore-like state in the membrane bound state. Here again RMSD and secondary structure evolution needs to be discussed.
7. Why did the authors not attempt to put three monomers in the membrane to study the structure in a membrane inserted pore state with hydrogen bonding between the beta-sheets? This would provide a lot of additional insights in comparison with the single inserted protein simulations which are not stabilized by surrounding proteins. I leave this as an optional simulation
8. The results and discussion section should be split into the different simulations. Membrane inserted state and the umbrella sampling simulations.
9. For Figure 2 simulations the authors need to plot the RMSD evolution with time as well as the secondary structure evolution with time and comment on these graphs (in SI). Figures should have sub-figure labels A, B ... and clearly described in the caption. Same point for Figure 3 and Figure 4 as well. Sub-figure labels should be used in the text of the manuscript.
10. Why was C191 used to define the reaction co-ordinate during the umbrella sampling simulations ? C191 should be marked in the Figure 3 for better clarity. Additionally the different values of zeta where snapshots have been shown should be marked in Figure for ease of following the different trends.
11. Page 7: This sentence "Then 207 snapshots for generating the potential of mean force (PMF) were taken from both the two equilibrium simulations in the inserted and in the adsorbed state and from the two pulling simulations (out and in) as well as from an additional simulation lasting 500 ns in which the z-distance between the C of C191 and the membrane was restrained to 0.5 nm to assure a great diversity of the simulation snapshots" is confusing at best. The authors should put these details along with a few snapshots to illustrate how the configurations (snapshots) or the umbrella sampling were

obtained. Due to the large barriers and the size of the protein it is likely that the pulling (out and in) configurations generated will be vastly different. These snapshots can be added to supplement Figure 1 in the supplementary information.

12. Usually pulling in causes large membrane perturbations even for smaller molecules in a membrane. What were the membrane structures during the pulling in simulations versus the pulling out simulations?

13. What is the crystal structure of the protein before it inserts into the membrane. In these simulations the authors assume that the protein structure of the pore state can be used to study the protein in the membrane bound state. This does not appear to be correct and should be discussed in the revised manuscript. Related to point 6 above.

14. PMF Computations: The authors should show how the PMF profiles change as the sampling is increased to 50 ns. 50 ns seem to be small for this large protein. Simulations must be carried out for at least 100 ns at each window for reliable discussion of the PMF. If not then the manner in which the PMF evolves from 20, 30, 40, 50 ns is needed. Additionally, usually the first part of the simulation at each window is discarded for better statistics. This point needs to be discussed and justified.

15. Additionally, secondary structure changes of the protein at a few windows during the umbrella sampling are needed to discuss the secondary structure evolution. This is particularly important for the membrane bound state as the structure determined by the membrane inserted pore state is not necessarily valid outside the membrane.

16. The authors mention that the GSDMA3 simulations have been carried out earlier. If so, appropriate acknowledgement and permissions are needed from the previous publisher before usage in this manuscript.

Reviewer #2

(Remarks to the Author)

The manuscript by Korn and Pluhackova describes results of molecular modelling of gasdermin monomers in contact with the membrane. Various gasdermins have important physiological roles and are structurally highly similar. Here, the authors show that there are notable differences in membrane insertion of two gasdermins, A3 and D, by performing extensive molecular simulations. The results build upon previous study on gasdermin A3 by the same authors. Hereby, they were complemented with data for gasdermin D and analyzed in relevance to the main questions asked in the study. The results are interesting for those working with gasdermins and potentially for researchers that study protein-membrane interactions.

The conclusions about differences in insertion efficiency between the two gasdermins could be supplemented with experimental results that would confirm the modelling results. The authors rightly include the published results (last paragraph of the Discussion), but the study could be complemented, for example, by a mutational study of residues highlighted in Figure 4.

In general, the figures would benefit from organization into clearly labeled panels.

The introduction section is very detailed. However, I find parts of the introduction not very useful or informative for this work, i.e. the extensive description of cleavage of gasdermins with caspases (lines 45-52) and this part could be drastically shortened.

Lines 29,30: The statement about gasdermins and their presence in different organisms requires citations from the literature.

Lines 75-77: two possible main pathways are mentioned here for the first time in the introduction. I suggest the rest of the paragraph is written differently, to briefly describe each of the models (as for example in done in the abstract) and cite appropriate references that propose each of the pathways.

Line 102: the exact composition of *E. coli* polar lipid extract membrane should be defined here.

Figure 1: The residue C191 should be marked in the sequence and on the structure in Figure 1. What are the side chains in structures? Why are some presented as balls and some as sticks? Additionally, PDB-IDs should be stated from which the structural models in Figure 1 were prepared.

Line 188: The differences (and similarities) of the both gasdermins should be presented in more detail at the beginning of the results section, i.e. the degree of similarity stated, how similar are structures, etc.

Line 196: I cannot find residue E171 in Figure 1 (sequence in the middle of the figure), please check labeling of residues.

Figure 2: Top right, the proteins used should be labeled. The details about engagement of phosphate groups and various amino acids in GSDMD are not visible and should be presented as an additional panel that would show the enlarged part of the structure with highlighted residues that mediate protein-lipid interactions.

Reviewer #3

(Remarks to the Author)

This study investigates the mechanisms of insertion of gasdermin D and gasdermin A3 using atomistic molecular dynamics simulations. The authors conclude that gasdermin D favors the membrane-inserted state and can insert into the membrane in a monomeric form, while gasdermin A3 prefers the membrane-adsorbed state, suggesting the need for oligomerization prior to membrane insertion. These results have potential implications for the pathways of pore formation. The text is well-

written and detailed. In the introduction, the authors provide extensive information on the categorization of gasdermins and their biological function but only lightly touch on membrane insertion, which is the main focus of the study. The methods section is very detailed and provides enough information to replicate the study. The simulation procedures used align with common practices.

1) The authors state that they based their model on the PDB structure 6VFE. This structure contains an engineered residue at position 192; could the authors clarify in the methods section whether this mutation was reverted?

2) Hysteresis is a common problem in free energy calculations (DOI: 10.1021/jp501622d or 10.1021/acs.jctc.6b00369). It is my understanding that the authors have used simulation snapshots from two pulling simulations (pull in and pull out, Figure 1 in the SI) to construct the PMF. Would it be possible to construct the PMF for each direction separately to get information on the hysteresis and the calculated error?

3) Could the authors show whether the insertion pathway is the same for both pull directions?

4) The authors state that each of the 207 simulations lasted only 50 ns. Commonly, the data points that are used to compute the profile are restricted, and the beginning of the simulation is discarded as the equilibration period. Could the authors clarify the frequency of the data point collection and if all the data points were used to compute the profile? Furthermore, would it be possible to show the change of the computed profiles in time to provide information on the profile convergence?

5) The reported values are roughly on the same scale as the transfer free energies of a single amino acid side-chain from water to the membrane interface (DOI: 10.1085/jgp.200709745). Given the limited simulation timescale and the size of the protein, could the authors comment on how much their conclusions would change if the protein assumed a different orientation on the membrane?

6) Is it meaningful to report the free energy barrier of membrane insertion (e.g., line 16) to two decimal places, given that the "standard deviations estimated over 4 random parts of the trajectories" in Figure 3 are more than 1 kcal/mol?

7) Could the authors comment on the choice of the collective variable and if the computed profiles and barriers are comparable? Given that "[...] FES can capture the relative stability of metastable states but not that of the transition state because the barrier height is not invariant to the choice of CVs. Free energy barriers therefore cannot be consistently computed from the FES." (DOI: 10.1063/5.0020240)

8) At line 223, the authors describe the protein-lipid interactions in detail. Given the complexity of the multi-component membrane, could the authors comment on whether the equilibration was sufficiently long for proper lipid mixing?

9) In the paragraph starting at line 298, the authors provide a detailed description of the formation of salt bridges. If this information is important to the reader, could the authors provide information on the dynamics of these interactions and clarify the meaning of the reported values? For example, if a salt bridge is formed for the first half of the simulation and then breaks apart, or if the salt bridge repeatedly forms and breaks and is formed for half of the simulation time in total. This would inform the reader about the simulation sampling and influence of the starting conditions.

Version 1:

Reviewer comments:

Reviewer #1

(Remarks to the Author)

The authors have addressed all the comments raised by the referees and the manuscript has been extensively revised and improved. It can be published without additional revisions.

Reviewer #2

(Remarks to the Author)

The authors have reformatted manuscript significantly and added new complementary data. My comments were addressed appropriately. I do not have further comments.

Reviewer #3

(Remarks to the Author)

I would like to thank the authors for addressing all of my comments. I appreciate the time and care they have taken in refining the manuscript. The revisions provide additional context and clarity, which enhances the overall readability of the work. The inclusion of additional figures is helpful in illustrating key points and increasing confidence in the robustness of the findings. At this point, I have no further comments.

We would like to sincerely thank the reviewers for their thorough evaluation of our manuscript and for their valuable feedback. We are grateful for their insightful and constructive comments, which have significantly contributed to improving the clarity and depth of our work towards both the expert and a broader, non-expert audience. Each remark and suggestion has helped us refine our analysis and presentation, and we have carefully addressed all the points raised to enhance the rigor and readability of our paper by restructuring the Introduction, the Methods section, as well as the results. In order to help introduce the non-specialist to the topic, we made two new explanatory schemes about pyroptosis and gasdermins' pore formation. The original Results section has been extended by a thorough analysis of the equilibrium simulations, including secondary structure analysis using RMSD and DSSP calculations. Moreover, we added simulations of GSDMD in a POPC/30% cholesterol bilayer, emphasizing the influence of the membrane lipid composition on interactions and pore formation dynamics. Taking a look on the broader picture, we supplemented our manuscript with oligomer simulations, as the reviewers cleverly suggested, to consolidate the trends we could observe for the monomer simulation.

In this rebuttal, we provide detailed responses to each of the reviewers' comments, explaining the revisions we have made in response to their observations. We appreciate the time and expertise that the reviewers have dedicated to evaluating our manuscript and trust that the improvements we have implemented align with their expectations.

Reviewer 1:

1. The introduction reads as one long paragraph. The authors need to split this into separate paragraphs and provide some pictures/schematics for the paper to be more accessible for the non-specialist.

We have split and rearranged the text in the introduction and also in the results section to increase the readability and furthermore added schematic graphics (Figure 1 and 2).

2. Figure 2: The numbers of the end residues should be marked in the respective CryoTEM structures for the reader to follow the sequence in the structure. The authors have superimposed the two structures to depict the pore. This is confusing. A separate and improved figure is needed to depict the pore for GSDMD along with the superposition to depict the differences.

We split the former Figure 2 into now Figure 3 depicting the sequences and Figure 4 depicting the overlay. Furthermore, labels of the first and last amino acid have been added and we highlighted C191 (GSDMD) and K97 (GSDMA3) which are used in the reaction coordinate.

3. These pores are usually formed with a range of monomers. The pore size being determined by the protein concentration, membrane composition etc.. This aspect should be mentioned unless there is clear evidence to the contrary

We added more information on different pore sizes:

"For both proteins, slit-shaped, elliptic and round pores with different numbers of subunits (D: 16, 20, 30,¹³ 26 to 28;¹² A3: 18, 21, 27, 30,¹⁴ 31 to 34⁴¹) have been reported. Lipid-filled gasdermin oligomers or oligomers adsorbed to the membrane surface are typically denoted as pre-pores which may be of different shapes (arc, slit, ring).^{13,14} Notably, the membrane's lipid composition has been proven to influence both the pore size and shape.^{2,89}"

And stressed the importance of the lipid composition in several places in the manuscript:

"The different behavior of GSDMD and GSDMA3 monomers is also reflected in the different pore-formation propensity of arcs comprising either seven GSDMD or seven GSDMA3 monomers. Moreover, simulations of monomeric GSDMD in POPC/30% cholesterol support the significant modulatory role of the membrane composition on gasdermin pore formation"

"While GSDMA3 maintains its arc shape even after 4 μ s of simulation time, the GSDMD oligomer pulls water molecules and lipid headgroups to its β -hairpins and wraps around forming an elliptic water-filled pore, similarly to its behavior in the plasma membrane observed earlier by Schäfer et al.⁴²"

"The importance of those specific protein-PE interactions is confirmed by our all-atom simulations of monomeric GSDMD inserted in a simple plasma-membrane mimic membrane consisting of POPC and 30% cholesterol. The lack of PE- β -hairpin interactions destabilizes the membrane-inserted state of GSDMD resulting in rapid repulsion of the β -hairpins from the membrane in two simulation and partial excision in the third simulation (see Supplementary Figures 10 and 11). "

" In order to further elaborate on the role of complex lipid composition for membrane pore formation, we have performed additional simulations of monomeric GSDMD inserted in a model membrane consisting of POPC with 30% cholesterol. Strikingly, the β -hairpins of GSDMD slipped out of the bilayer within a few μ s in two out of three simulations (see Supplementary Figure 10 and 11), proving the modulatory role of specific lipids in stabilizing GSDMD in its membrane-inserted state. Moreover, the differences of GSDMD and GSDMA3 monomers concerning the formation of water-filled membrane defects and pores are a foundation for the vastly different behavior of their larger oligomers. Here we have studied arcs comprised of seven monomers. While the arc of GSDMA3 stays stable over the course of the simulation and causes only small water defects, the arc of GSDMD pulls water and lipid headgroups to its β -hairpins, forming a free membrane edge. The generated edge tension of the membrane rim pulls the terminal monomers of GSDMD together, wrapping it around the water defect, and forming a water-filled transmembrane pore, similarly to the behavior of GSDMD in the plasma membrane model.⁴² "

4. It was not clear how the water channel is stabilized when the pore forms. Typically for the beta-toxins from the family of cholesterol dependent cytolysins the hydrophilic regions form the water channel with the hydrophobic regions facing the lipids. Is there any similarity in this trend for the proteins studied here. If so this should be mentioned somewhere to draw a connect. Hence it is not entirely obvious if longer beta-hairpins should lead to a higher barrier for entry.

To clarify this point, we added the information concerning hydrophobic and hydrophilic parts and the water channel to the first Results section:

"In the full pores of GSDMD and A3, the inside is more hydrophilic and the outside, facing the lipids, more hydrophobic,⁴¹ similar to other β -barrel forming proteins^{88,89}. Acidic patches line the inside of the pore and basic ones the outside facing the lipids. Thereby GSDMD exhibits a stronger electrostatic surface.^{12,41} Thus, based on the primary protein structure of the GSDMD monomer, one could expect a higher barrier due to the longer β -hairpins that bear more hydrophilic and charged residues than for GSDMA3, as inserting charged residues into a hydrophobic membrane core comes at a high energy cost.^{90,94}"

5. A table of the different simulations carried out should be mentioned in the methods section. Again this section should be split up into the restraint simulations and the umbrella sampling simulations.

We moved the table from the Supplementary Information to the main text (Table 1) and extended it with more details. We also split descriptions of equilibrium simulations and umbrella sampling into different sections.

In the Results and Discussion

6. In the methods section the authors mention that they carry out simulations with the membrane bound states. However this data is not shown. It would be useful to comment on the differences observed for the two proteins when compared with the membrane inserted states. This would complement the membrane inserted states show in Figure 2. These simulations will also reveal the stability fo the pore-like state in the membrane bound state. Here again RMSD and secondary structure evolution needs to be discussed.

We added figures (Figure 5) and text about the equilibrium simulations in both the inserted and the membrane bound states:

" (see Figure 5c and Supplementary Figure 5b). This behavior is also reflected in the RMSD (see Supplementary Figure 6), where the hairpins' RMSD deviates from the pore structure by 0.2 to 0.6 nm, with maximum values reached for inserted GSDMD at around 1 μ s. As the secondary structure analysis (see Supplementary Figure 7) does not show any significant loss of β -sheet residues, these increased RMSD values reflect reorientation of the whole hairpins rather than their unfolding. The globular domain remains stable both in the inserted and the adsorbed state (Supplementary Figure 6, a, b)."

and plotted and analyzed the RMSD, zeta, and the secondary structure content (Supplementary Figures 6, 7, 10).

7. Why did the authors not attempt to put three monomers in the membrane to study the structure in a membrane inserted pore state with hydrogen bonding between the beta-sheets? This would provide a lot of additional insights in comparison with the single inserted protein simulations which are not stabilized by surrounding proteins. I leave this as an optional simulation

We have added 7mer simulations (seven to be comparable to GSDMA3 arc behavior, published recently). (Table 1, Figure 7)

"The above observed differences of monomeric GSDMD and GSDMA3 in their ability to form water-filled membrane pores gets even more pronounced if multiple proteins are membrane inserted as shown in our simulations of membrane inserted arcs consisting either of seven GSDMD or seven GSDMA3 monomers (see Figure 7). While GSDMA3 maintains its arc shape even after 4 μ s of simulation time, the GSDMD oligomer pulls water molecules and lipid headgroups to its β -hairpins and wraps around forming an elliptic water-filled pore, similarly to its behavior in the plasma membrane observed earlier by Schäfer et al.⁴² "

8. The results and discussion section should be split into the different simulations. Membrane inserted state and the umbrella sampling simulations.

As stated above, everything has been split and rearranged.

9. For Figure 2 simulations the authors need to plot the RMSD evolution with time as well as the secondary structure evolution with time and comment on these graphs (in SI). Figures should have sub-figure labels A, B ... and clearly described in the caption. Same point for Figure 3 and Figure 4 as well. Sub-figure labels should be used in the text of the manuscript.

We added subfigure labels to the figures and added the RMSD and DSSP analyses to the supplementary information (Supplementary Figure 6 and 7).

10. Why was C191 used to define the reaction co-ordinate during the umbrella sampling simulations ? C191 should be marked in the Figure 3 for better clarity. Additionally the different values of zeta where snapshots have been shown should be marked in Figure for ease of following the different trends.

C191 was highlighted in Figure 3 together with K97

"Highlighted in red are C191 (GSDMD) and K97 (GSDMA3), which were selected as part of the reaction coordinate for geometric perturbation simulations."

"C191 is characteristic for GSDMD and due to its location at the tip of the β -hairpin (Figure 3) makes the reaction coordinate comparable to that of GSDMA3, where the distance between K97, also located at the tip of the β -hairpin, and the membrane middle along the membrane normal was selected as the reaction coordinate."

11. Page 7: This sentence "Then 207 snapshots for generating the potential of mean force (PMF) were taken from both the two equilibrium simulations in the inserted and in the adsorbed state and from the two pulling simulations (out and in) as well as from an additional simulation lasting 500 ns in which the z-distance between the C of C191 and the membrane was restrained to 0.5 nm to assure a great diversity of the simulation snapshots" is confusing at best. The authors should put these details along with a few snapshots to illustrate how the configurations (snapshots) or the umbrella sampling were obtained. Due the large barriers and the size of the protein it is likely that the pulling (out and in) configurations generated will be vastly different. These snapshots can be added to supplement Figure 1 in the supplementary information.

We extended Table 1 and corrected Supplementary Figure 1 to clarify the origin of the snapshots. Additionally, Supplementary Figure 3 shows exemplary pictures from the pull in and pull out simulations of frames that were used for geometric perturbation. Using frames from different simulations, we want to ensure that the entire reaction coordinate is covered and different configurations level out hysteresis. The confusing sentence has been rewritten to:

" 207 snapshots were taken from (i) the equilibrium simulation in the inserted state, (ii) the equilibrium simulation in the adsorbed state, (iii) the pull-out simulation, (iv) the pull-in simulation and, (v) a simulation lasting 500 ns in which the z-distance between the C_{α} of C191 and the membrane was restrained to 0.5 nm. See Table 1 and Supplementary Figure 1 and 2 for further details and Supplementary Figure 3 for exemplary snapshots."

12. Usually pulling in causes large membrane perturbations even for smaller molecules in a membrane. What were the membrane structures during the pulling in simulations versus the pulling out simulations ?

We added pictures in Supplementary Figure 3. We made sure by visually inspecting the pulling simulations that only frames with an intact membrane were used for geometric perturbation. On top of that, many structures were pre-equilibrated additionally (see Supplementary Figure 2 description).

13. What is the crystal structure of the protein before it inserts into the membrane. In these simulations the authors assume that the protein structure of the pore state can be used to study the protein in the membrane bound state. This does not appear to be correct and should be discussed in the revised manuscript. Related to point 6 above.

We are aware that the protein structure in the adsorbed state is not fully equilibrated and likely doesn't represent the native conformation. Unfortunately, the current computational resources do not allow us to simulate β -sheet folding in the adsorbed state or upon membrane insertion. On the other hand, the assumption that the protein structure similar to that of the pore state may be used to initiate the protein insertion allows a direct comparison to the insertion of GSDMA3. We have commented on this problem in the Results section:

" Keeping in mind the limited structural sampling (folding/unfolding) of the β -hairpins of both GSDMD and GSDMA3 in the cytosol, the comparison of the simulations shows that the membrane-adsorbed state of GSDMD is not characterized by a single potential well as in the case of GSDMA3 but comprises multiple small potential wells"

Additionally, we plotted the RMSDs for different protein parts in the equilibrium simulations in comparison to the crystal structures (Supplementary Figure 6) and analyzed the protein's secondary structure content.

14. PMF Computations: The authors should show how the PMF profiles changes as the sampling is increased to 50 ns. 50 ns seem to small for this large protein. Simulations must be carried out for at least 100 ns at each window for reliable discussion of the PMF. If not then the manner in which the PMF evolves from 20, 30, 40 50 ns is needed. Additionally usually the first part of the simulation at each window is discarded for better statistics. This point needs to be discussed and justified.

We have added the requested profiles (Supplementary Figure 4) which clearly show that a sampling time of 50 ns is sufficient. On top of that, many structures were equilibrated additionally (see Supplementary Figure 2 description).

15. Additionally secondary structure changes of the protein at a few windows during the umbrella sampling is needed to discuss about the secondary structure evolution. This is particularly important for the membrane bound state as the structure determined by the membrane inserted pore state is not necessarily valid outside the membrane.

For the umbrella sampling in the membrane adsorbed state, we were using frames from an equilibrium simulation in the adsorbed state and fro the inserted part, we used frames from an inserted equilibrium simulation (Table 1, Supplementary Figure 1). We also did DSSP calculations for all umbrella and equilibrium simulations (Supplementary Figure 7) which show that no major unfolding events are happening.

"As the secondary structure analysis (see Supplementary Figure 7) does not show any significant loss of β -sheet residues, these increased RMSD values reflect reorientation of the whole hairpins rather than their unfolding. The globular domain remains stable both in the inserted and the adsorbed state (Supplementary Figure 6, a, b)."

16. The authors mention that the GSDMA3 simulations have been carried out earlier. If so appropriate acknowledgement and permissions are needed from the previous publisher before usage in this manuscript.

We double checked and it's all legitimate. The original publication of the GSDMA3 simulations was published under open access license. Therefore, citation of the data is sufficient.

Reviewer 2:

The conclusions about differences in insertion efficiency between the two gasdermins could be supplemented with experimental results that would confirm the modelling results. The authors rightly include the published results (last paragraph of the Discussion), but the study could be complemented, for example, by mutational study of residues highlighted in Figure 4

Unfortunately, we are a theoretical/computational lab and don't have the resources for such experiments.

In general, the figures would benefit from organization into clearly labeled panels.

We have added subfigure labels.

The introduction section is very detailed. However, I find parts of the introduction not very useful or informative for this work, i.e. the extensive description of cleavage of gasdermins with caspases (lines 45-52) and this part could be drastically shortened.

The caspase part has been omitted and everything else restructured.

Lines 29,30: The statement about gasdermins and their presence in different organisms requires citations from the literature.

Citations have been added.

" certain bacteria, ¹⁻³ fungi, ³⁻⁵ and invertebrates ^{3,6}"

Lines 75-77: two possible main pathways are mentioned here for the first time in the introduction. I suggest the rest of the paragraph is written differently, to briefly describe each of the models (as for example in done in the abstract) and cite appropriate references that propose each of the pathways.

We added clarifying text and references as well as an explanatory figure (Figure 2).

" For pore-forming proteins localized initially in solution like lysins, perforins, gasdermins, and Bcl-2 proteins, ⁵¹ two possible main pathways of pore formation have been proposed: The "concerted" pathway in which after activation, the proteins oligomerize and adsorb to the membrane prior to their simultaneous insertion, and the "non-concerted" one where after activation, monomers adsorb and insert individually before oligomerizing and forming the pore. ⁵¹ For visualization see Figure 2. Pneumolysin follows the concerted mechanism while actinoporin EqtII follows the non-concerted

one.⁵¹ For GSDMs, the data from experiments using atomic force microscopy^{13,14} and extensive molecular dynamics (MD) simulations,^{2,14,42,52} is consistent with both suggested pathways."

Line 102: the exact composition of E. coli polar lipid extract membrane should be defined here.

The information has been added to the methods section:

" The E. coli PLE membrane model consisted of 14 different lipid types: four cardiolipins, five phosphatidylglycerols, and five phosphatidylethanolamines, each including five different lipid tails: palmitoyl (16:0), palmitoleic acid (16:1 cis^{9,10}), cis-11,12- octadecenoic-acid (18:1 cis^{11,12}), cis-9,10-methylene-hexadecanoic-acid (cy17:0 cis^{9,10}), and cis-11,12-methylene-octadecanoic-acid (cy19:0 cis^{11,12}).⁵⁷ The lipids were symmetrically distributed between the two membrane leaflets."

Figure 1: The residue C191 should be marked in the sequence and on the structure in Figure 1. What are the side chains in structures? Why are some presented as balls and some as sticks? Additionally, PDB-IDs should be stated from which the structural models in Figure 1 were prepared.

C191 has been highlighted in now Figure 3, PDB-IDs have been added to Figures 3 and 4, explanations for representations have been added to Figure 3.

Line 188: The differences (and similarities) of the both gasdermins should be presented in more detail at the beginning of the results section, i.e. the degree of similarity stated, how similar are structures, etc.

In the Results section, we added:

" Despite the conserved overall secondary structure, the sequences of different gasdermins diverge notably, also introducing local distinctions like β -hairpins of different length and polarity. While the different length of the transmembrane β -hairpins could be essential for pore formation in membranes of different thickness, their amino acid composition influences interactions and energetics with the lipids and water to a level where they may regulate the type of pore-forming mechanism"

and Figures 3 and 4.

"The GSDMD monomer possesses longer β -hairpins than GSDMA3 (see Figure 4) and a characteristic C191 residue at one tip (Figure 3) which is known to play an important role during GSDMD pore formation.^{2,38} Next to the length also the amino acid sequence of the β -hairpins differs significantly. GSDMD carries a manifold of polar and charged residues across the entire hairpins in comparison to GSDMA3 (Figure 3), which only bears charged residues on its lower and upper end of the hairpins."

Line 196: I cannot find residue E171 in Figure 1 (sequence in the middle of the figure), please check labeling of residues.

Rebuttal Figure 1: Position of E171 (shown as green licorice, see arrow) on GSDMD (green cartoon) shown relative to the membrane phosphates within 1 nm of the protein in the last 1 μ s of the equilibrium simulation in the inserted state.

E171 does not have lipid contact in any of our simulations. Therefore, it is not included in Figure 1, where only membrane-interacting residues are shown. We highlight this selection in the legend of Figure 3 as:

"CryoTEM structures of GSDMD (green) (6VFE) and GSDMA3 (navy) (6CB8) monomers in comparison. On the membrane-inserting β -hairpins, polar residues are shown as cartoon and charged residues shown as ball&stick and colored purple (positively charged lysine and arginine) and yellow (negatively charged aspartic and glutamic acid) together with their amino acid sequences. Highlighted in red are C191 (GSDMD) and K97 (GSDMA3), whose position relative to the membrane's center of mass constitute the reaction coordinate for geometric perturbation simulations."

Furthermore, we discuss the position of E171 in the main text:

" In detail, while D87 and E95 at AP2 are important for the specificity of the cargo release, substitution of all the glutamic acids E169, E171, and E179 at AP3 for alanine compromised GSDMD-mediated liposome leakage of membranes made of cardiolipin, phosphatidylethanolamine lipids and phosphatidylcholine lipids mixed in 5:8:4 ratio. Notably, E171 and E169 are located above the lipid bilayer and do not interact with lipids in our simulations. Therefore it is likely E179 which is responsible for the lipid specificity."

Figure 2: Top right, the proteins used should be labeled. The details about engagement of phosphate groups and various amino acids in GSDMD are not visible and should be presented as an additional panel that would show the enlarged part of the structure with highlighted residues that mediate protein-lipid interactions.

Subfigure labels were added, the figure has been split, and labels have been added. Figure 6 now depicts the phosphate groups and interacting amino acids more clearly.

Reviewer 3:

1) The authors state that they based their model on the PDB structure 6VFE. This structure contains an engineered residue at position 192; could the authors clarify in the methods section whether this mutation was reverted?

The mutation from 6VFE has been reversed and a sentence stating this has been added to the beginning of the Methods section:

"The structure of GSDMD in the membrane-inserted state was based on the cryoTEM structure 6VFE 12 with the E192 mutation reversed to L192..."

2) Hysteresis is a common problem in free energy calculations (DOI: 10.1021/jp501622d or 10.1021/acs.jctc.6b00369). It is my understanding that the authors have used simulation snapshots from two pulling simulations (pull in and pull out, Figure 1 in the SI) to construct the PMF. Would it be possible to construct the PMF for each direction separately to get information on the hysteresis and the calculated error?

We agree with the reviewer that hysteresis is a common problem in free energy calculations and we appreciate his suggestion. Unfortunately, it is not easily possible to reasonably separate the simulation snapshots used for determination of the GSDMD's PMF, as we do not only use snapshots from pulling in and pulling out but also from inserted, adsorbed and "restrained-in the bilayer" state (see Supplementary Figure 1). This makes the definition of the "there" and "back" path for the hysteresis ambiguous. Also, for GSDMA3 we did not pull the protein but extracted one fraction of the umbrellas from the part of the trajectory in which GSDMA3 slipped out of the bilayer and complemented it by frames from the inserted and adsorbed states, which makes it impossible to evaluate the hysteresis effect there. Therefore, instead of evaluating the hysteresis we prefer to base our conclusions on a single PMF of GSDMD insertion/excision based on snapshots of different origin, which is visible in Supplementary Figure 1 showing that critical ζ area from -2 to 2 nm is always being covered by snapshots of at least two different origins.

3) Could the authors show whether the insertion pathway is the same for both pull directions?

Unfortunately we cannot as the snapshots used from pulling in and pulling out do only partially overlap and thus do not cover the exact same ζ (reaction coordinate) range, as it is depicted in Supplementary Figure 1. In Supplementary Figure 3, we included graphical depictions of some snapshots from the pulling-in and pulling-out simulations, which show different configurations of the β -hairpins of GSDMD.

4) The authors state that each of the 207 simulations lasted only 50 ns. Commonly, the data points that are used to compute the profile are restricted, and the beginning of the simulation is discarded as the equilibration period. Could the authors clarify the frequency of the data point collection and if all the data points were used to compute the profile? Furthermore, would it be possible to show the change of the computed profiles in time to provide information on the profile convergence?

We added a figure (Supplementary Figure 4) which shows the convergence of the PMFs. As it can be seen in this Figure, 50 ns of simulation time for each geometric perturbation simulation is sufficient. We used all the data points for the Gromacs tool WHAM and did not discard anything (otherwise we would have stated such in the Methods section), as many configurations stem from equilibrium simulations (see Main Text Table 1 and Supplementary Figure 1). Additionally, a large fraction of the frames from the pulling simulations underwent an additional equilibration before running the actual geometric perturbation simulation (Supplementary Figure 2). Some snapshots had to be taken as-is to ensure proper coverage of the reaction coordinate (Supplementary Figure 2).

5) The reported values are roughly on the same scale as the transfer free energies of a single amino acid side-chain from water to the membrane interface (DOI: 10.1085/jgp.200709745). Given the limited simulation timescale and the size of the protein, could the authors comment on how much their conclusions would change if the protein assumed a different orientation on the membrane?

Already in our first work on the membrane insertion/excision of GSDMA3 we found it striking that the effects (discussed in the manuscript in the Results and Discussion) including piggy back, salt bridge formation, snorkeling, lipid bending, and water defect formation lead to the PMF profile which gives similar values to the transfer free energies of single amino acids or their side chain analogs. Our results and literature search agree however on the cooperativity effect of the amino acids in a protein environment. We agree that if the hairpins have to fold first or upon the membrane insertion that the shape of the PMF could change, however, it is so far not feasible to reach the computational times (ms timescale for β -sheet unfolding/folding) required for such observations. Our main goal is to compare the PMFs of insertion/excision of GSDMA3 and GSDMD into the same lipid composition and point out the similarities and differences. To highlight this, we have included a following comment in the manuscript:

" Keeping in mind the limited structural sampling (folding/unfolding) of the β -hairpins of both GSDMD and GSDMA3 in the cytosol, the comparison of the simulations shows that the membrane-adsorbed state of GSDMD is not characterized by a single potential well as in the case of GSDMA3 but comprises multiple small potential wells."

6) Is it meaningful to report the free energy barrier of membrane insertion (e.g., line 16) to two decimal places, given that the "standard deviations estimated over 4 random parts of the trajectories" in Figure 3 are more than 1 kcal/mol?

We discarded the second decimal for all results.

7) Could the authors comment on the choice of the collective variable and if the computed profiles and barriers are comparable? Given that "[...] FES can capture the relative stability of metastable states but not that of the transition state because the barrier height is not invariant to the choice of CVs. Free energy barriers therefore cannot be consistently computed from the FES." (DOI: 10.1063/5.0020240)

As it can be seen in our updated Figure 3, both C191 (GSDMD) and K97 (GSDMA3) are located at the tip of the β -hairpins and allow a comparison between the two proteins. Furthermore, C191 is characteristic for GSDMD:

"...and a characteristic C191 residue at one tip (Figure 3) which is known to play an important role during GSDMD pore formation. 2,38"

Selecting the C_{α} of these two amino acids as an "anchor" allows the rest of the protein to be flexible and adapt different conformations which is especially important regarding slight differences between the adsorbed and inserted states (see Supplementary Figures 5 and 6). The same goes for the z-direction of the membrane normal: All other directions stay flexible and unrestricted, allowing the membrane to assume a more natural conformation. We do not claim to have calculated absolute free energies, which is never possible anyways, but rather stress the differences between the relative free energies of insertion/excision and behaviors of two proteins from the same protein family.

8) At line 223, the authors describe the protein-lipid interactions in detail. Given the complexity of the multi-component membrane, could the authors comment on whether the equilibration was sufficiently long for proper lipid mixing?

Rebuttal Figure 2: Left. Gasdermin D in E. coli PLE at 0 μ s. Right. Gasdermin D in E. coli PLE after 5 μ s in the equilibrium simulation. The protein is shown as green cartoon, the lipid phosphates are grey except for 7 phosphates forming, together with the gasdermin's hairpins, a water-filled pore at 5 μ s. Those are colored red, both at 5 μ s and at 0 μ s.

As it can be seen in the above figure, the lipids are well mixed and none of the defect-lining ones were near the protein at the beginning of the simulation, which makes sense regarding the use of a pre-equilibrated membrane and an additional 5 μ s of equilibration time.

9) In the paragraph starting at line 298, the authors provide a detailed description of the formation of salt bridges. If this information is important to the reader, could the authors provide information on the dynamics of these interactions and clarify the meaning of the reported values? For example, if a salt bridge is formed for the first half of the simulation and then breaks apart, or if the salt bridge repeatedly forms and breaks and is formed for half of the simulation time in total. This would inform the reader about the simulation sampling and influence of the starting conditions.

We have included exemplary time courses of the salt bridges for GSDMD at $\zeta = -1.5, 0,$ and 1.5 nm in Supplementary Figure 13.

There also is a paragraph in the Main text describing them, which we now extended even further to include more details:

" As salt bridge formation is an important mechanism shielding off charges on the protein surface from the membrane environment,^{52,103–106} we have compared the salt bridge formation capability of GSDMA3 and GSDMD over the course of the geometric perturbation simulations. While GSDMA3 possesses just four charged amino acids at the β -hairpins that are spatially able to form salt bridges (E94, K97, K100, and K102) and only two of them do actually form one (K100E94, see Figure 9b,d), GSDMD carries a number of charged residues on its β -hairpins, theoretically allowing the formation of 18 salt bridges, of which seven are actually formed in our simulations (Figure 9a,c). Exemplary time courses are shown in Supplementary Figures 13–15. K103 mostly interacts with E95 (69.7 % of total simulation time), sometimes it gets closer to E195 (5.6 % of total simulation time) ($-1.2 < \zeta < 0.3$), which mostly interacts with R183 (39.7 % of total simulation time). Sometimes R183 can be seen switching over to E95 (8.9 % of total simulation time) (mostly between $\zeta = -1.2$ and 0.3), so K103 and R183 share their partners. R174 mostly interacts with D112 (87.4 % of total simulation time), forming the most stable and continuous salt bridge. K177 mostly forms a salt bridge with E179 (39.2 % of

total simulation time). Only during early insertion, it briefly interacts with D87 (7.8 % of total simulation time) ($-1.7 < \zeta < 0.2$), too. Generally, the salt bridges are present in the cytosol, too, and fluctuate more inside the membrane."